# Atomic-level insight into super-efficient electrocatalytic oxygen evolution on iron and vanadium co-doped nickel (oxy)hydroxide

Jian Jiang[1], Fanfei Sun[2], Si Zhou[3], Wei Hu[4], Hao Zhang[2], Jinchao Dong[5], Zheng Jiang [2], Jijun Zhao[3], Jianfeng Li [5], Wensheng Yan[4] & Mei Wang[1]

It is of great importance to understand the origin of high oxygen-evolving activity of state-of-the-art multimetal oxides/(oxy)hydroxides at atomic level. Herein we report an evident improvement of oxygen evolution reaction activity via incorporating iron and vanadium into nickel hydroxide lattices. X-ray photoelectron/absorption spectroscopies reveal the synergistic interaction between iron/vanadium dopants and nickel in the host matrix, which subtly modulates local coordination environments and electronic structures of the iron/vanadium/nickel cations. Further, in-situ X-ray absorption spectroscopic analyses manifest contraction of metal–oxygen bond lengths in the activated catalyst, with a short vanadium–oxygen bond distance. Density functional theory calculations indicate that the vanadium site of the iron/vanadium co-doped nickel (oxy)hydroxide gives near-optimal binding energies of oxygen evolution reaction intermediates and has lower overpotential compared with nickel and iron sites. These findings suggest that the doped vanadium with distorted geometric and disturbed electronic structures makes crucial contribution to high activity of the trimetallic catalyst.

[1] State Key Laboratory of Fine Chemicals, DUT-KTH Joint Education and Research Centre on Molecular Devices, Dalian University of Technology, Dalian 116024, China. [2] Shanghai Institute of Applied Physics, Chinese Academy of Sciences, Shanghai 201204, China. [3] MOE Key Laboratory of Materials Modification by Laser, Ion and Electron Beams, Dalian University of Technology, Dalian 116024, China. [4] National Synchrotron Radiation Laboratory, University of Science and Technology of China, Hefei 230029, China. [5] State Key Laboratory of Physical Chemistry of Solid Surfaces, College of Chemistry and Chemical Engineering, Xiamen University, Xiamen 361005, China. These authors contributed equally: Jian Jiang, Fanfei Sun. Correspondence and requests for materials should be addressed to W.Y. (email: ywsh2000@ustc.edu.cn) or to M.W. (email: symbueno@dlut.edu.cn)

Hydrogen as an energy-dense and carbon-neutral fuel is an ideal alternative to the limited fossil fuels to sustain the fast increase in energy consumption by human society. Water splitting to hydrogen and oxygen ($H_2O \rightarrow H_2 + 1/2O_2$), driven by electric power generated from renewable energy sources, is known as a promising approach to hydrogen production in a large-scale[1,2]. To this end, one of the crucial challenges is to develop inexpensive electrocatalysts that are highly active and durable for water oxidation and proton reduction.

Among the reported non-noble metal catalysts for oxygen evolution reaction (OER), Ni-based bimetal oxides[3–8], especially NiFe layered-double-hydroxides (LDHs) (refs. [9–16]), have drawn intensive attention due to their excellent OER performance in alkaline media. Much recent research revealed that the incorporation of a third transition metal into NiFe oxides/hydroxides to form NiFeM (M = Co (refs. [17–19]), Mn (ref. [20]), Cr (refs. [21,22]), and Al (refs. [23,24])) ternary composites could further enhance the intrinsic OER activity of the Ni–Fe (oxy)hydroxide catalyst in different extents[25]. In another aspect, the unary vanadium (oxy)hydroxide was demonstrated to be a highly active OER electrocatalyst in alkaline solution[26]. Some very recent studies discovered that incorporation of V into Ni- or/and Fe-based oxides/(oxy)hydroxides could effectively enhance the OER activity of the catalysts[27–30]. However, the questions remain on whether V has substitutionally doped into the lattices of host materials and if so, how V dopant interplays with other metal ions co-existing in a catalyst material, and how the doped V cations contribute to the high OER activity of the host materials. To our knowledge, to date, there is no report on in-depth spectroscopic studies of local coordination environments and electronic structures for the V-containing bi- and trimetal (oxy)hydroxide OER catalysts in both rest and activated states. In very recent years, several groups made in-depth studies on NiFe (refs. [10,14,31–34]), CoFe (ref. [35]), NiFeCo (ref. [36]), FeCoW (ref. [37]), and NiFeCoCe (ref. [38]) oxides/(oxy)hydroxides by employing X-ray absorption spectroscopy (XAS), especially operando XAS measured during electrolysis of a catalyst at a preset applied potential. The results obtained from these significant studies provided some crucial information for understanding the origin of high activity of these catalysts and for identifying the authentic active sites in the catalysts.

In light of the reports mentioned above, we prepared a series of Fe/V co-doped, Fe- or V-doped, and pure Ni (oxy)hydroxides as ultrathin nanosheets (NSs) on hydrophilic carbon fiber paper (CFP), and made comparative studies on these OER catalysts by X-ray photoelectron spectroscopy (XPS) and ex-situ/in-situ XAS, combined with density functional theory (DFT) calculations. The Fourier and wavelet transform (FT/WT) analyses of the extended X-ray absorption fine structure (EXAFS) data demonstrate the substitutional occupation sites of Fe and V dopants in $Ni(OH)_2$ lattices, consistent with the results obtained from theoretical calculations. Moreover, XPS and XAS analyses reveal the synergetic interaction of Fe, V, and Ni cations in the $Ni_3Fe_{0.5}V_{0.5}$ catalyst, which subtly modulates local coordination environments and electronic structures of Ni/Fe/V cations in the catalyst. Further in-situ XAS studies manifest a different extent of shrinkage of metal–oxygen bond lengths in the activated catalyst, with a short V–O1 bond distance of 1.65 Å. DFT calculations indicate that the V site of the Fe/V co-doped Ni (oxy)hydroxide gives near-optimal binding energies (BEs) of OER intermediates, and point to the higher OER activity of V site compared to that of Ni and Fe sites.

## Results

### Fabrication and characterization of $Ni_3Fe_{1-x}V_x$.
A series of $Ni_3Fe_{1-x}V_x$ (oxy)hydroxide catalysts ($0 \leq x \leq 1$), namely $Ni_3Fe$,

$Ni_3V$, $Ni_3Fe_{0.67}V_{0.33}$, $Ni_3Fe_{0.5}V_{0.5}$, $Ni_3Fe_{0.33}V_{0.67}$, and pure Ni (oxy)hydroxides were directly grown on hydrophilic CFPs by hydrothermal synthesis (Fig. 1 and Supplementary Fig. 1). The atomic ratios of different metals in the as-prepared catalysts were determined by analyses of inductively coupled plasma optical emission spectroscopy (ICP-OES, Supplementary Table 1).

The powder X-ray diffraction (PXRD) patterns (Supplementary Fig. 2) indicate that $Ni_3Fe_{1-x}V_x$ are isostructural to $\alpha$-Ni$(OH)_2$ (JCPDS Card No. 38-0715). The reflections at $2\theta = 11.4°$ and 22.7°, corresponding to the (003) and (006) lattice planes of $Ni_3Fe_{1-x}V_x$, slightly shift to larger $2\theta$ values by 0.2° and 0.6°, respectively, relative to those of $\alpha$-Ni$(OH)_2$. The $d$-spacing values obtained from the (003) and (006) reflections are about 7.65 and 3.81 Å, respectively, with a small contraction compared to those for pure $\alpha$-Ni$(OH)_2$ ($d(003) = 7.79$ Å and $d(006) = 3.91$ Å), which is most possibly caused by the substitution of Fe and V atoms for Ni in the lattice sites of the $Ni(OH)_2$ matrix[32,39,40]. Because no extra diffraction peaks are observed in the PXRD pattern, it could be deduced that no separated crystalline phases, such as unary Ni-, Fe-, or V-based oxides/(oxy)hydroxides, are formed during the doping process[12,28,41].

Scanning electron microscopic (SEM) images of $Ni_3Fe_{0.5}V_{0.5}$/CFP (Fig. 2a) clearly show that the entire surface of each carbon fiber is uniformly coated with the densely interlaced NSs, forming a sharp contrast to the smooth surface of pristine carbon fibers (Supplementary Fig. 3). A close inspection (Fig. 2b) reveals that the interlaced NSs form a porous network structure. Such an open nanoarchitecture built by $Ni_3Fe_{0.5}V_{0.5}$ ultrathin NSs would afford a mass of electrochemically active sites, an easy penetration of electrolyte, and a good mechanical strength, so as to improve the OER activity and stability of the electrodes[4]. The quantitative SEM energy dispersive X-ray spectrum (SEM-EDX) of $Ni_3Fe_{0.5}V_{0.5}$/CFP discloses the presence of Ni, Fe, V, and O elements with a Ni/Fe/V atomic ratio of 74.12:12.83:13.05, which is close to the stoichiometric metal ratio of Ni/Fe/V = 3:0.5:0.5. Moreover, the corresponding elemental mappings (Supplementary Fig. 4) illustrate that the Ni, Fe, V, and O elements distribute homogenously on the surface of carbon fibers.

The bright-field TEM (BF-TEM) image (Fig. 2c) of $Ni_3Fe_{0.5}V_{0.5}$ NSs illustrates a rippled sheet structure with a dimension around 500 nm, and the lateral TEM image (Fig. 2d) shows the ultrathin $Ni_3Fe_{0.5}V_{0.5}$ NSs with the thickness of 2.7–4.2 nm. Furthermore, the atomic-resolution BF-TEM image (Fig. 2e) displays clear lattice fringes with an interplanar spacing of 2.67 Å, indexed to the (101) plane of $Ni_3Fe_{0.5}V_{0.5}$ NSs. The interplanar spacing of lattice fringes is slightly smaller than that of $\alpha$-Ni$(OH)_2$ (2.68 Å) due to the doping of Fe and V for Ni in $Ni(OH)_2$ lattices. Single atoms, clusters, and small particles of Fe and V species are not observed in aberration-corrected high-angle annular dark-field scanning TEM (HAADF-STEM) images of $Ni_3Fe_{0.5}V_{0.5}$ NSs (Supplementary Fig. 5). Meanwhile, both the EDX elemental mappings and linear scanning analysis of the HAADF-STEM image of $Ni_3Fe_{0.5}V_{0.5}$ NSs with sub-nanometer resolution (Fig. 2f and Supplementary Fig. 6) provide direct-viewing evidence for the uniform distribution of Ni, Fe, V, and O elements in the as-prepared NSs.

In order to clarify the occupation sites of Fe and V dopants in $Ni(OH)_2$ lattices, we display in Fig. 3 the FT curves of the Fe and V $K$-edge EXAFS $k^2\chi(k)$ functions for $Ni_3Fe$, $Ni_3V$, and $Ni_3Fe_{0.5}V_{0.5}$. As references, their Ni $K$-edge FT curves are also plotted (Fig. 3a). The FT curves of the Fe $K$-edge data of $Ni_3Fe$ and $Ni_3Fe_{0.5}V_{0.5}$ exhibit two prominent coordination peaks at 1.5 and 2.7 Å that are identical to those of their Ni $K$-edge data (Fig. 3b), suggesting the substitutional doping of Fe in the Ni$(OH)_2$ host. Similarly, the FT curves of $Ni_3V$ and $Ni_3Fe_{0.5}V_{0.5}$ each display a prominent V–O peak at 1.4 Å and a V–M (M = Fe,

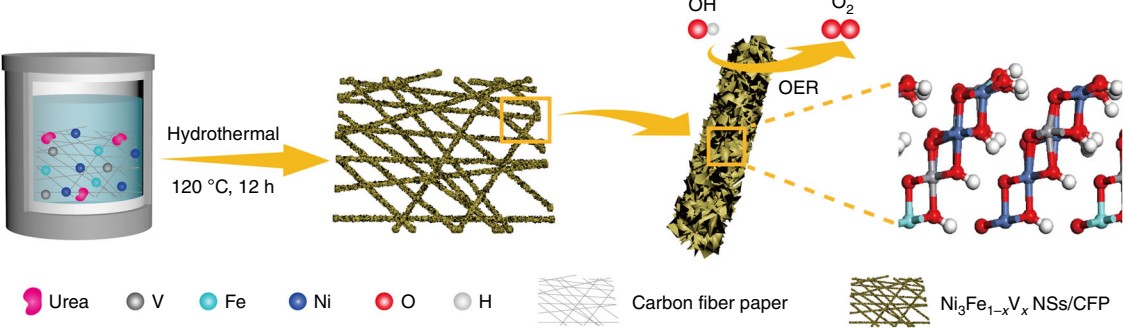

**Fig. 1** Fabrication of $Ni_3Fe_{1-x}V_x$/CFP $O_2$-evolving electrodes. Schematic illustration of the fabrication procedure by directly growing $Ni_3Fe_{1-x}V_x$ NSs on a pretreated CFP substrate

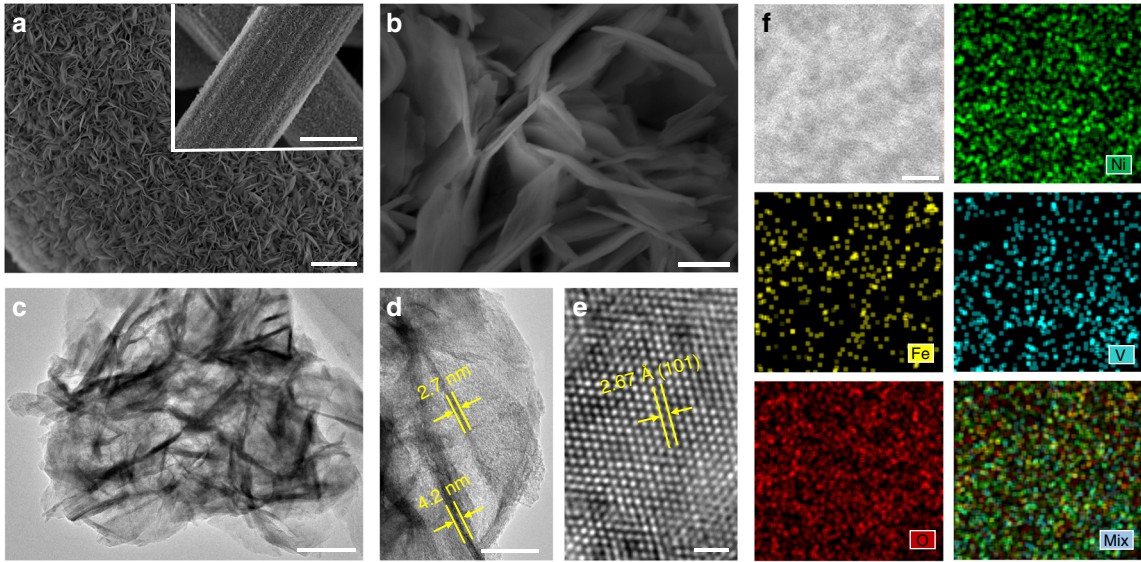

**Fig. 2** Microscopy measurements of $Ni_3Fe_{0.5}V_{0.5}$ NSs. **a**, **b** Top-view SEM images of the carbon fiber coated with $Ni_3Fe_{0.5}V_{0.5}$ NSs with low (**a**) and high (**b**) magnification. Scale bars, 1 μm in **a** and 100 nm in **b**. The inset in **a** shows the hierarchically structured 3D integrated electrode. Scale bar in the inset in **a**, 5 μm. **c**, **d** TEM images of $Ni_3Fe_{0.5}V_{0.5}$ NSs scratched off from the as-prepared CFP electrode. Scale bars, 100 nm in **c** and 40 nm in **d.e** Atomic-resolution BF-TEM image of $Ni_3Fe_{0.5}V_{0.5}$ NSs. Scale bar, 1 nm. **f** Aberration-corrected HAADF-STEM image of $Ni_3Fe_{0.5}V_{0.5}$ NSs, the corresponding EDX elemental mappings of Ni, Fe, V, O and the mixed elemental mapping of Ni, Fe, V, and O. Scale bar, 2 nm

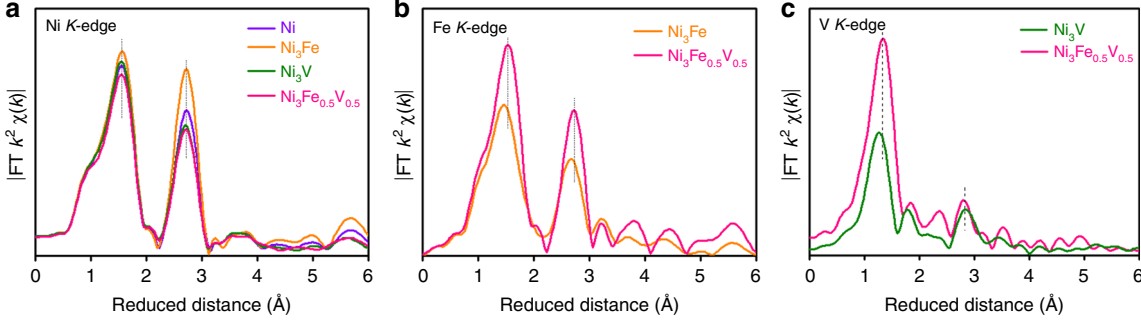

**Fig. 3** XAS spectra of the as-prepared Ni-based (oxy)hydroxide catalysts. FT curves of **a** Ni K-edge, **b** Fe K-edge, and **c** V K-edge EXAFS $k^2\chi(k)$ functions

Ni, or V) peak at about 2.8 Å (Fig. 3c), and the high-shell peak of $Ni_3Fe_{0.5}V_{0.5}$ is weaker than that of $Ni_3V$. The significant decrease in the intensity of the V–M coordination peak in the FT curve of $Ni_3Fe_{0.5}V_{0.5}$ is most likely caused by the highly distorted local structure of V substituting for the site of Ni. To confirm the

substitution of V for Ni in $Ni(OH)_2$ lattices, the WT analysis of the V K-edge data was performed. A maximum at the cross point of R = 2.8 Å and k = 7.8 Å$^{-1}$ appears in the EXAFS WT map at the V K-edge for $Ni_3Fe_{0.5}V_{0.5}$ (Supplementary Fig. 7), just like that for $Ni_3V$. This implies the presence of V–Fe/Ni scatterings at

a distance of around 2.8 Å surrounding V atoms and affords direct evidence for the substitution of V atoms for the Ni sites in $Ni(OH)_2$ lattices. We also made the calculation of the EXAFS spectra by assuming V adsorption on the Ni–Fe LDH layer or occupying the interstitial position. It turns out that in both cases the calculated spectra are quite different from the experimental V $K$-edge EXAFS spectra of $Ni_3Fe_{0.5}V_{0.5}$ (Supplementary Fig. 8). Furthermore, DFT calculations suggest that V atoms initially placed on the top site of surface Ni or O atoms are relaxed to the interstitial between two LDH layers after structure optimization. The LDH structure with interstitial doping is noticeably buckled, with formation energy of −3.73 eV per V atom, less stable with regard to LDH with substitutional doping (−5.07 eV per V atom) (Supplementary Figs. 9, 10), supporting that V atoms occupy Ni positions in $Ni(OH)_2$ lattices rather than the interstitial or top positions of LDH layers. On the other side, from Supplementary Fig. 11, the nearest-neighbor FT peak position of V is shifted to the lower-$R$ side and the second coordination peak to higher-$R$ side with apparently reduced intensity as compared to that of Fe. This implies the remarkable different local environment of the substitutional V from that of Fe in $Ni_3Fe_{0.5}V_{0.5}$. The quantitative parameters extracted from EXAFS curve-fitting (Supplementary Figs. 12–14 and Supplementary Tables 2–4) further show that the bond length of V–O (1.72 Å) is significantly contracted with regard to those of Fe–O (2.00 Å) and Ni–O (2.03 Å).

**Understanding the electronic interaction in $Ni_3Fe_{1−x}V_x$.** The electronic states of Fe and V in catalysts were investigated by ex-situ hard X-ray absorption near-edge spectroscopy (XANES). Generally, in XANES spectra the intensity of the pre-edge peak depends predominantly on central site symmetry, while the absorption edge position is correlated to the oxidation state of central sites[42]. The absorption edges of $Ni_3Fe$, $Ni_3V$, and $Ni_3Fe_{0.5}V_{0.5}$ in the XANES curves of Ni $K$-edge (Supplementary Fig. 15a) are all alike to that of the original $Ni(OH)_2$, indicative of nearly identical average oxidation states of Ni in the catalysts. Similarly, the XANES curves of Fe $K$-edge in Supplementary Fig. 15b show that the adsorption edges of Fe for $Ni_3Fe$, $Ni_3Fe_{0.5}V_{0.5}$, and $Fe_2O_3$ reference are almost overlapped, manifesting that the average valence states of Fe are close to +3 in the as-prepared catalysts. Importantly, the V $K$-edge XANES spectra of $Ni_3V$ and $Ni_3Fe_{0.5}V_{0.5}$ exhibit intense pre-edge peaks (Supplementary Fig. 15c), indicating the distorted coordination environment around V atoms in these materials[42]. More interestingly, $Ni_3Fe_{0.5}V_{0.5}$ shows a higher pre-edge peak than that of $Ni_3V$ in the V $K$-edge XANES, implying a higher degree of octahedral geometry distortion at the V sites in $Ni_3Fe_{0.5}V_{0.5}$ compared to those in $Ni_3V$. Additionally, the $K$-edge absorption positions of $Ni_3V$ and $Ni_3Fe_{0.5}V_{0.5}$ are more close to those of $VO_2$ and $V_2O_5$ than to that of $V_2O_3$ (inset of Supplementary Fig. 15c), suggesting that the majority of V ions are in the formal valences of +4 and +5 in both catalysts.

The as-prepared $Ni_3Fe_{1−x}V_x$ films were further studied by XPS and ex-situ soft XAS to gain an insight into the electronic interaction between Fe/V dopants and Ni atoms at the surface of catalysts. For $Ni_3Fe_{0.5}V_{0.5}$ NSs, the Ni $2p$ spectrum (Fig. 4a) exhibits two fitting peaks at 872.3 and 854.4 eV along with two shakeup satellites at 878.4 and 860.1 eV, which are characteristic spin-orbit peaks of $Ni^{2+}$ (refs. [13,28,43]). In the Fe $2p$ region (Fig. 4b), Fe $2p_{1/2}$ and Fe $2p_{3/2}$ peaks arise at 724.8 and 711.5 eV, indicative of Fe in the +3 oxidation state (refs. [16,29]). The V $2p_{3/2}$ peak (Fig. 4c) can be deconvoluted into three peaks located at 516.2 eV ($V^{5+}$), 515.1 eV ($V^{4+}$), and 514.4 eV ($V^{3+}$) (refs. [26,28,29]), demonstrating that the V atoms are predominantly in high oxidation states (+4 and +5) in $Ni_3Fe_{0.5}V_{0.5}$, together

with a minority of $V^{3+}$, which is consistent with the results obtained from V $K$-edge XANES spectra.

It is worthy of note that the Ni $2p$ BEs for the Fe or/and V doped binary and ternary materials are shifted apparently to higher BEs compared to those of pure Ni (oxy)hydroxide, with the shift extent in an increasing order of $Ni_3Fe < Ni_3V < Ni_3Fe_{0.5}V_{0.5}$ (Fig. 4a, Supplementary Fig. 16a, and Supplementary Table 5). In contrast, the V $2p$ peaks for $Ni_3V$ are shifted to lower BEs relative to the corresponding peaks for $VO_2$ (ref. [44]), and $Ni_3Fe_{0.5}V_{0.5}$ displays V $2p_{3/2}$ peaks at BEs ~0.2 eV lower than those of $Ni_3V$ (Fig. 4c and Supplementary Fig. 16c). Of particular interest is that the BEs of Fe $2p_{1/2}$ and $2p_{3/2}$ for $Ni_3Fe$ are lower than those for $Fe_2O_3$ (ref. [45]), but when half amount of Fe in $Ni_3Fe$ is replaced by V, $Ni_3Fe_{0.5}V_{0.5}$ exhibits Fe $2p$ peaks at BEs not only considerably higher than those of $Ni_3Fe$ but also higher than $Fe_2O_3$ (Fig. 4b and Supplementary Fig. 16b), implying that the Fe dopant acts as an electron accepting site in $Ni_3Fe$ but an electron donating site in an integrated effect when V is co-doped with Fe into $Ni(OH)_2$ lattices. These observations suggest the partial electron transfer from Ni to Fe or V in the bimetal (oxy) hydroxides through oxygen bridges ($O^{2−}$) between metal ions, and from Ni and Fe to V in $Ni_3Fe_{0.5}V_{0.5}$, which is in good agreement with the calculated Mulliken charges for V, Fe, and Ni ions in $Ni_3Fe_{1−x}V_x$ (Supplementary Table 6).

These speculations are further supported by the Ni, Fe, and V $L$-edge XANES spectra shown in Fig. 4d–f. Figure 4d illustrates that doping Fe or V could intensify the Ni $L_3$-edge peak (852.5 eV), indicative of partial electron transfer from Ni to the substitutional Fe or V. The intensity of the Fe $L_3$-edge peak at 709.8 eV for $Ni_3Fe$ is also enhanced when V is doped into $Ni_3Fe$ (Fig. 4e). On the contrary, the V $L_3$-edge peak (518.3 eV) of $Ni_3Fe_{0.5}V_{0.5}$ is considerably weakened and shows a red-shift, as compared with that of $Ni_3V$ (Fig. 4f). The comparative analyses of XPS and XANES spectra suggest that co-doping of Fe together with V into $Ni(OH)_2$ lattices results in more electron transfer to the V in $Ni_3Fe_{0.5}V_{0.5}$ compared to that in $Ni_3V$ (Fig. 4c, f and Supplementary Fig. 16c). The strong interaction among these $3d$ metal ions results in synergistic modulation of the electronic structure of the metal centers of Fe/V co-doped $Ni(OH)_2$ (refs. [12,22,36,43]), and the concerted effect of Ni, Fe, and V metals with different energy levels of $d$-band centers could make crucial contribution to the evident enhancement of OER activity of hybridized materials. Moreover, we calculated the branching ratio, $L_3/(L_2 + L_3)$, at the Fe $L$-edges of $Ni_3Fe$ and $Ni_3Fe_{0.5}V_{0.5}$, which is approximately 0.74, implying the high-spin of $Fe^{3+}$ (ref. [46]). And we also calculated the Fe $L_{2,3}$-edge XAS for the high-spin and low-spin models of $Fe^{3+}$ (Supplementary Note 1 and Supplementary Methods). Obviously, the calculated high-spin $L_{2,3}$-edge XAS could well produce the experimental data (Supplementary Fig. 17), affording more evidence for the high-spin configuration of $Fe^{3+}$ substituting the Ni sites. Thus, the valence electronic configurations of $Ni^{2+}$, $Fe^{3+}$, $V^{4+}$, and $V^{5+}$ are $3d^8$ ($t_{2g}^6e_g^2$), $3d^5$ ($t_{2g}^3e_g^2$), $3d^1$ ($t_{2g}^1e_g^0$), and $3d^0$ ($t_{2g}^0e_g^0$), respectively, which are adopted in the following analysis of valence electron structures of metal ions in $Ni_3Fe$, $Ni_3V$, and $Ni_3Fe_{0.5}V_{0.5}$.

The synergistically electronic interplay of Ni, Fe, and V cations in $Ni_3Fe_{0.5}V_{0.5}$ is well explained in light of the analysis of valence electron structures of metal ions. In term of the result obtained from DFT calculations that the $(Ni_3Fe_{0.5}V_{0.5})$-OOH models with some aggregated Fe and V atoms have lower formation energy and higher OER activity than the models with isolated Fe and V atoms (vide infra), a Ni–O–Fe–O–V–O–Ni unit (Fig. 4g) is used to analyze the electronic interaction of Ni, Fe, and V cations in $Ni_3Fe_{0.5}V_{0.5}$. For $Ni_3Fe$, the three unpaired electrons in the $\pi$-symmetry ($t_{2g}$) $d$-orbitals of $Fe^{3+}$ interplay with the

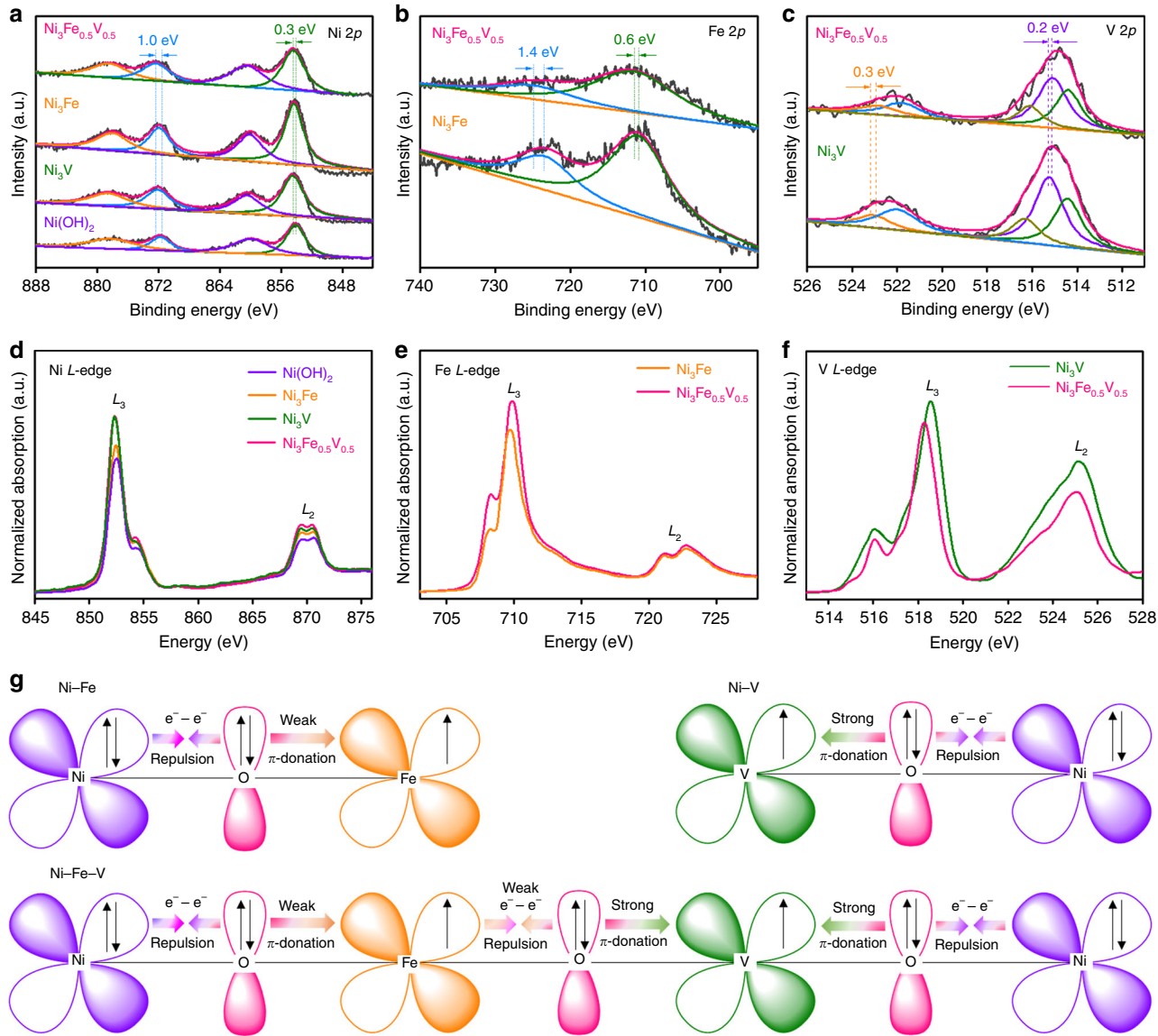

**Fig. 4** High-resolution XPS and XANES spectra of the as-prepared Ni-based (oxy)hydroxide catalysts. XP spectra of **a** Ni 2$p$ for Ni$_3$Fe$_{0.5}$V$_{0.5}$, Ni$_3$Fe, Ni$_3$V, and pure Ni (oxy)hydroxides, **b** Fe 2$p$ for Ni$_3$Fe and Ni$_3$Fe$_{0.5}$V$_{0.5}$, and **c** V 2$p$ for Ni$_3$V and Ni$_3$Fe$_{0.5}$V$_{0.5}$ (the dashed lines shown in **a**–**c** indicate the shifts of BEs of 3$d$ metal ions caused by the hybridization of Fe or/and V dopants). **d**–**f** Ni, Fe, and V $L$-edge XANES spectra. **g** Schematic representations of the electronic coupling among Ni, Fe, and V in Ni$_3$Fe, Ni$_3$V, and Ni$_3$Fe$_{0.5}$V$_{0.5}$

bridging $O^{2-}$ via $\pi$-donation, while the dominant interaction between the fully occupied $\pi$-symmetry ($t_{2g}$) $d$-orbitals of Ni$^{2+}$ and the bridging $O^{2-}$ is electron–electron repulsion, leading to partial electron transfer from Ni$^{2+}$ to Fe$^{3+}$ (refs. [12,40]). The partial electron transfer from $\pi$-symmetry lone pairs of the bridging $O^{2-}$ to V$^{4+}$ and V$^{5+}$ in Ni$_3$V should be stronger than that from the bridging $O^{2-}$ to Fe$^{3+}$ in Ni$_3$Fe, as V$^{4+}$ and V$^{5+}$ have rather low $t_{2g}$ occupancy while Fe$^{3+}$ has a half $t_{2g}$ occupancy. As for the Fe/V co-doped Ni(OH)$_2$ with some of the V and Fe atoms aggregated in the host lattices, when Fe$^{3+}$ accepts partial electrons from Ni$^{2+}$ through the bridging $O^{2-}$ via $\pi$-donation as exampled by the NiFe (oxy)hydroxide reference, the electron-riched $t_{2g}$ $d$-orbitals of Fe$^{3+}$ could relay electrons to the strongly electron-deficient $t_{2g}$ $d$-orbitals of V$^{4+}$ and V$^{5+}$ through the bridging $O^{2-}$ ions between them, which leads to better delocalization of the $\pi$-symmetry electrons among Ni, Fe, and V in the host matrix. This argument is in good agreement with the XPS and soft XANES results. In Ni$_3$Fe$_{1-x}$V$_x$, the Fe$^{3+}$

and Ni$^{2+}$ with half-full $e_g$ orbitals would have very weak bonding with adsorbed oxygen species, whereas the V$^{4+}$ and V$^{5+}$ with $e_g^0$ orbitals would form too strong bonding with adsorbed oxygen species. To get high OER activity, the bonding strength between transition metal and adsorbed oxygen species should be optimized to fulfill the Sabatier principle[47]. With increase of the electron density on V by partial electron transfer from Fe and Ni to V through the bridging $O^{2-}$ ions, the high valence states of V could be stabilized under OER conditions, and more importantly, the strong bond strength between V and adsorbed oxygen species could be tuned to a moderate bond strength, which would benefit for releasing O$_2$ from the V site in OER.

**Evaluating the electrochemical OER performance of Ni$_3$Fe$_{1-x}$V$_x$.** The electrocatalytic OER performance of Ni$_3$Fe$_{1-x}$V$_x$/CFP were studied in O$_2$-saturated 1 M KOH. The linear sweep voltammograms (LSVs, Fig. 5a) of all as-prepared Ni-based

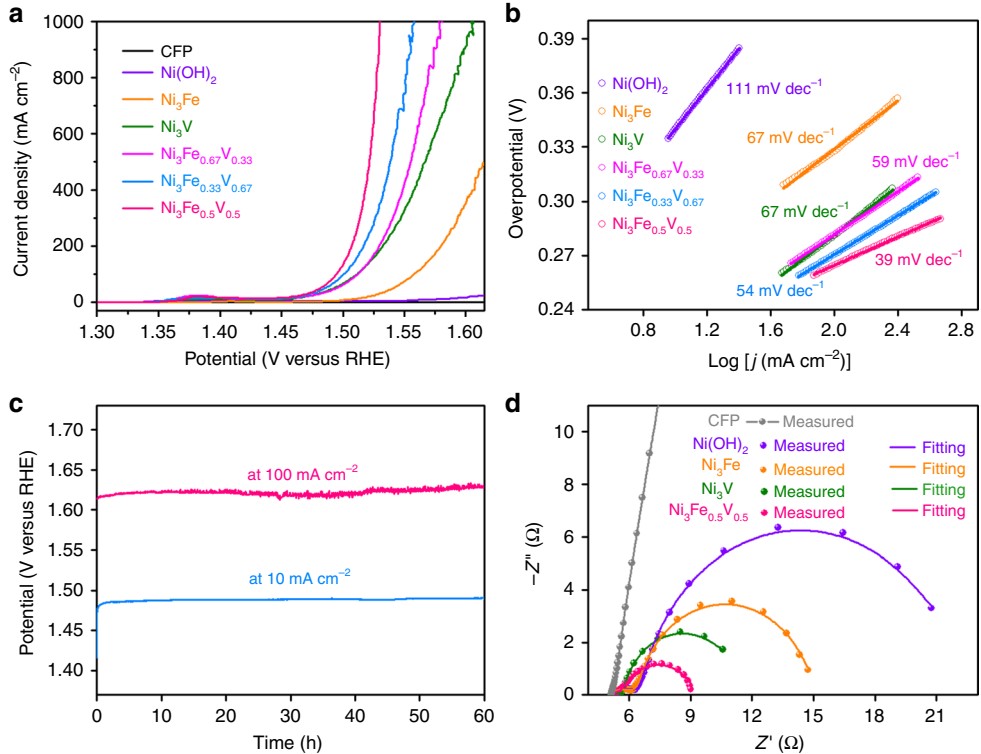

**Fig. 5** Electrochemical tests for OER and Nyquist plots of the Ni-based (oxy)hydroxide catalysts. **a** LSV curves of $Ni_3Fe_{1-x}V_x$ and pure Ni (oxy) hydroxide catalysts on CFP, as well as bare CFP in $O_2$-saturated 1 M KOH at a scan rate of 5 mV s$^{-1}$. **b** Tafel plots derived from the polarization curves in **a**. **c** Chronopotentiometric curves obtained with $Ni_3Fe_{0.5}V_{0.5}$ at constant current densities of 10 and 100 mA cm$^{-2}$. **d** Nyquist plots of $Ni_3Fe_{0.5}V_{0.5}$, $Ni_3Fe$, $Ni_3V$, and Ni (oxy)hydroxides and the bare CFP at 300 mV overpotential in 1 M KOH

(oxy)hydroxide catalysts show the Ni$^{2+}$/Ni$^{3+}$ oxidation in the potential range of 1.33–1.42 V (all potentials are versus reversible hydrogen electrode (RHE))[5,7,43].

Figure 5a illustrates that the electrocatalytic activity of $Ni_3Fe_1$ $_{-x}V_x$ depends largely on the co-doping level of Fe and V atoms. Among the as-prepared Fe- or/and V-doped Ni-based binary and ternary catalysts, $Ni_3Fe_{0.5}V_{0.5}$ exhibits the best OER performance, with low overpotentials of 264 and 291 mV to achieve 100 and 500 mA cm$^{-2}$ current density, respectively (Supplementary Fig. 18). The LSV of $Ni_3Fe_{0.5}V_{0.5}$, scanning from positive to negative direction to exclude the influence of the Ni$^{2+}$/Ni$^{3+}$ oxidation event on the catalytic current, shows that only 200 mV overpotential is required to attain 10 mA cm$^{-2}$ current density. The OER performance of $Ni_3Fe_{0.5}V_{0.5}$ is on a par with or even surpasses that of the first-class earth-abundant catalysts reported to date (Supplementary Table 7).

Moreover, the turnover frequency (TOF, based on total amount of metals) of $Ni_3Fe_{0.5}V_{0.5}$ (0.574 s$^{-1}$) at $\eta = 300$ mV in 1 M KOH is significantly larger than those of $Ni_3Fe$ (0.018 s$^{-1}$), $Ni_3V$ (0.097 s$^{-1}$), $Ni_3Fe_{0.67}V_{0.33}$ (0.116 s$^{-1}$), and $Ni_3Fe_{0.33}V_{0.67}$ (0.195 s$^{-1}$). Figure 5b manifests that the Tafel slope of $Ni_3Fe_{0.5}V_{0.5}$ (39 mV dec$^{-1}$) is considerably smaller than those of $Ni_3Fe_{0.67}V_{0.33}$ (59 mV dec$^{-1}$), $Ni_3Fe_{0.33}V_{0.67}$ (54 mV dec$^{-1}$), $Ni_3Fe$ (67 mV dec$^{-1}$), and $Ni_3V$ (67 mV dec$^{-1}$). The apparently larger TOF value and smaller Tafel slope of $Ni_3Fe_{0.5}V_{0.5}$ as compared to those of $Ni_3Fe$ and $Ni_3V$ indicate that the synergetic effect of co-doped Fe and V plays an important role in facilitating the kinetics of OER and enhancing the intrinsic activity.

The stability of $Ni_3Fe_{0.5}V_{0.5}$ was assessed by repeated cyclic voltammetry scanning, multi-current step test, and long-term chronopotentiometric experiments. After being subjected to 4000 CV cycles, the OER polarization curve of $Ni_3Fe_{0.5}V_{0.5}$ almost overlaps with the initial one (Supplementary Fig. 20a), indicating

no noticeable loss in catalytic current, and thus, the good accelerated stability of the electrode. Supplementary Fig. 20b shows the E–t plot of two cycles of multi-current step curves for $Ni_3Fe_{0.5}V_{0.5}$ with current density being enhanced from 50 to 500 mA cm$^{-2}$ by five steps. In each step, once a certain current density is set, the potential promptly levels off and maintains constant for 500 s; the multi-current step curve is well repeated in the subsequent cycle. This observation signifies fast mass transportation and good electronic conductivity of the 3D $Ni_3Fe_{0.5}V_{0.5}$/CFP matrix[13]. Additionally, the $Ni_3Fe_{0.5}V_{0.5}$/CFP electrode displays good stability at fixed current densities of 10 and 100 mA cm$^{-2}$, respectively, over 60 h of electrolysis (Fig. 5c), indicating excellent stability of the electrode under testing conditions. The Faradaic efficiency of $Ni_3Fe_{0.5}V_{0.5}$ evaluated from a chronopotentiometric experiment at a constant current density of 10 mA cm$^{-2}$ for 2 h is close to 100% (Supplementary Fig. 21).

To have a general understanding on the superior activity of $Ni_3Fe_{0.5}V_{0.5}$, we estimated the roughness factors (RF) and measured the electrochemical impedance spectroscopy (EIS) of all as-prepared Ni-based electrodes. Based on the estimated RF values (Supplementary Fig. 22 and Supplementary Note 2), the OER polarization curves of $Ni_3Fe_{1-x}V_x$ were ploted with $J$ normalized by RF values (Supplementary Fig. 23) to estimate the improvement of intrinsic OER activity for the Ni-based (oxy) hydroxides with different Fe and V doping levels[3]. The specific current density ($J_s = 61.6$ mA cm$^{-2}$) of $Ni_3Fe_{0.5}V_{0.5}$ at 300 mV overpotential is about 3, 16, and 71 times higher than those of $Ni_3V$, $Ni_3Fe$, and pure Ni (oxy)hydroxides, respectively, which reveals that the co-doping of Fe and V into Ni(OH)$_2$ lattices is much more effective than separately doping Fe or V for improving the specific activity of Ni-based catalysts, and the improved specific activity contributed largely to the high OER

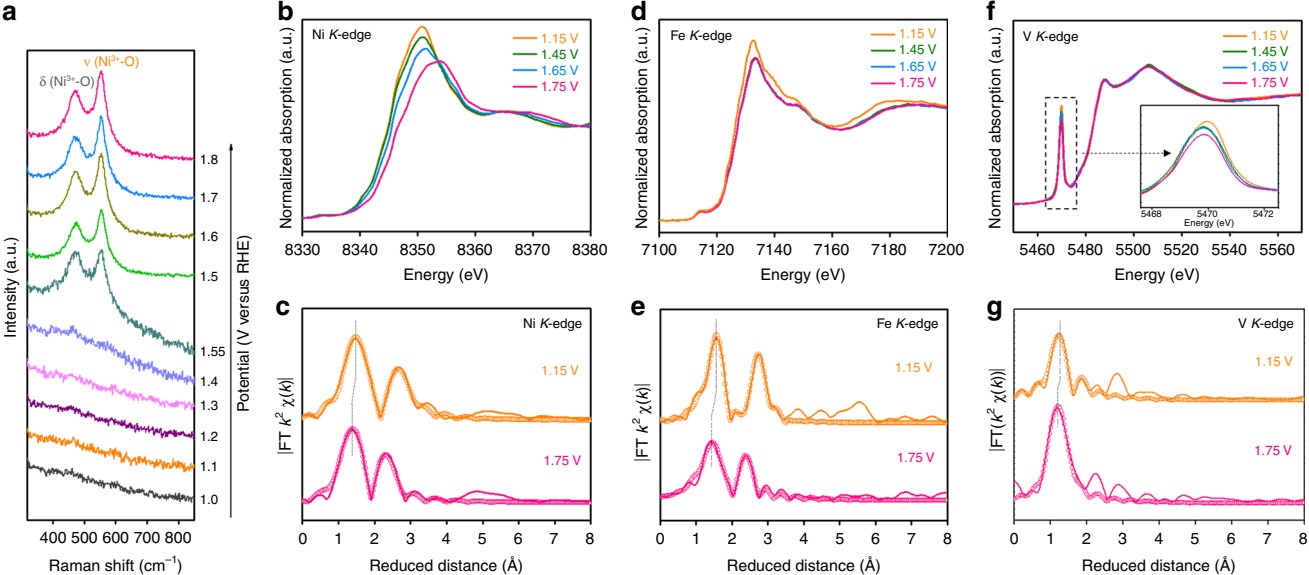

**Fig. 6** In-situ EC-Raman and hard XAS spectra. **a** In-situ EC-Raman spectra of $Ni_3Fe_{0.5}V_{0.5}$ at the potentials of 1.0–1.8 V in 1 M KOH. **b, d, f** In-situ Ni, Fe, and V K-edge XANES spectra at the potentials of 1.15–1.75 V. **c, e, g** FT curves of Ni, Fe, and V K-edge EXAFS $k^2\chi(k)$ functions obtained from the XANES spectra in **b, d, f**, respectively. The orange and pink circles represent the fitting values

activity of $Ni_3Fe_{0.5}V_{0.5}$. The Nyquist plots (Fig. 5d) are fitted to a simplified Randles equivalent circuit model (Supplementary Note 3). The very small semicircles in the high frequency zone are attributed to the internal charge-transfer resistances ($R_{ct(int)}$) of electrodes, and the second semicircles represent the charge-transfer resistances ($R_{ct(s-l)}$) at the electrode/electrolyte interface. Both $R_{ct(int)}$ and $R_{ct(s-l)}$ values apparently decreased as Fe and V were co-doped into $Ni(OH)_2$ lattices. The total charge-transfer resistances ($R_{ct}$) measured at 300 mV overpotential are 4.2, 7.2, 10.0, and 17.2 Ω for the CFP-supported $Ni_3Fe_{0.5}V_{0.5}$, $Ni_3V$, $Ni_3Fe$, and pure Ni (oxy)hydroxides, respectively (Supplementary Table 9). The excellent charge-transfer capability of $Ni_3Fe_{0.5}V_{0.5}$ makes a crucial contribution to the superior intrinsic OER activity of the electrode[48].

**In-situ EC-Raman/XAS studies and theoretical calculations**. To have an in-depth insight into the origin of high activity of the Fe/V co-doped $Ni(OH)_2$, the changes in electronic structures and local atomic environments of $Ni_3Fe_{0.5}V_{0.5}$ under OER conditions were studied by in-situ electrochemical Raman (EC-Raman) spectroscopy and in-situ XAS. The measurements of in-situ EC-Raman spectra were carried out at the potential range of 1.0–1.8 V in a spectroelectrochemical (PEC) cell filled with 1 M KOH electrolyte (Fig. 6a). When the applied potential was higher than 1.4 V, a pair of well-defined Raman peaks at around 470 and 550 $cm^{-1}$ appeared, which were correlated respectively with the $e_g$ bending and the $A_{1g}$ stretching vibration of Ni–O in the NiOOH-type phase[49,50]. On the basis of EC-Raman spectra, the host phase of $Ni^{III}$–OOH, formed during the OER process, could provide an electrically conductive, chemically stable, and electrolyte-permeable framework for the Fe and V dopants[48,51], which would benefit the electrochemical OER.

Furthermore, the alteration in the local coordination environment of Ni–O/Fe–O/V–O units and the average oxidation states of Ni, Fe, and V centers in $Ni_3Fe_{0.5}V_{0.5}$ were investigated by in-situ hard XAS (Supplementary Fig. 24). The in-situ Ni K-edge XANES spectra (Fig. 6b) show that the Ni absorption-edge and the white-line are gradually shifted to the higher-energy side as the applied potential is increased from 1.15 to 1.75 V. Accordingly, the Ni–O distance is shortened from 2.04 Å at 1.15 V to

1.90 Å at 1.75 V (Fig. 6c and Supplementary Table 8). The former is close to the Ni–O bond length (2.05 Å) in $Ni(OH)_2$, and the latter is almost identical with the Ni–O bond length (1.88 Å) in NiOOH, which contains a mixture of $Ni^{3+}$ and $Ni^{4+}$ sites[10]. This is in line with the results of in-situ EC-Raman spectroscopy. A similar shift of the Fe white-line peak toward the higher-energy side is also observed with increasing applied potential (Fig. 6d), and the Fe–O distance is shortened slightly from 2.00 Å at 1.15 V to 1.97 Å at 1.75 V (Fig. 6e and Supplementary Table 8), signifying that the oxidation state of Fe is increased from +3 to nearly +4 during the OER electrolysis process of $Ni_3Fe_{0.5}V_{0.5}$. These FT-EXAFS fit results of Ni and Fe K-edges of $Ni_3Fe_{0.5}V_{0.5}$ in both rest and activated states are consistent with the previous reports[10,14]. More interestingly, with increasing the applied potential from 1.15 to 1.75 V, the pre-edge peak is slightly decreased in intensity in the in-situ V K-edge XANES spectrum (inset of Fig. 6f), however, it shows identical spectral features to those measured before OER. Similarly, except for the decrease in the intensity of the characteristic peaks, no other obvious change is visible at the ex-situ V L-edge spectra (Supplementary Fig. 25) after OER measurement at 1.75 V. This evidence suggests partial electron transfer to the V 3d orbitals, as their peak intensity is proportional to the unoccupied density of 3d states. Meanwhile, the V–O1 distance is also shrunk from 1.70 Å at 1.15 V to 1.65 Å at 1.75 V (Fig. 6g and Supplementary Table 8), which is close to that of the shortest V–O bond length reported for $V^{5+}$ oxides (1.59 Å) while much shorter than that reported for $V^{4+}$ oxides (1.76 Å)[42]. The V atoms with such a short V–O bond may have optimal binding capability with oxygen intermediates relative to Ni and Fe atoms, and exhibit enhanced OER activity, as will be illustrated by following theoretical calculations. These in-situ XAS analyses manifest for the first time the contraction of M–M′ and M(M′)–O bond lengths and the short V–O bond distance in the activated V-containing (oxy)hydroxide OER catalysts.

DFT plus Hubbard U (DFT + U) calculations were conducted to have a theoretical understanding on the evident enhancement of OER activity of the Fe/V co-doped $Ni(OH)_2$ from atomic level. It is known that $Ni(OH)_2$ experiences phase transformations during charging and discharging, and its oxyhydroxides are proposed to be the active phase for OER[14,52,53]. Thus, we

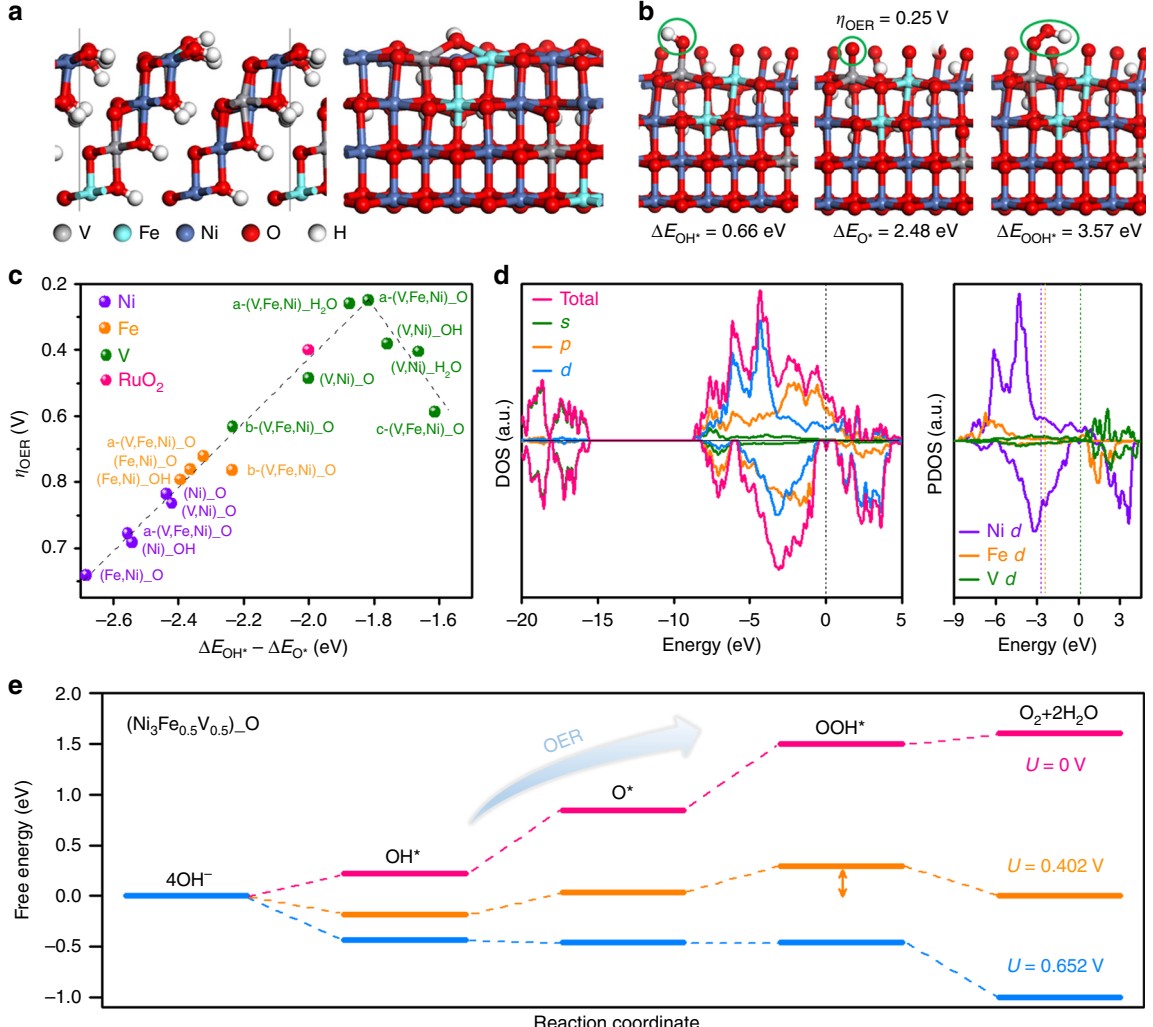

**Fig. 7** DFT theoretical models. **a** Side views of the Fe/V co-doped Ni (oxy)hydroxide model for DFT calculations, whose (101) surface is exposed for OER catalysis. **b** Structures and BEs of an OH*, O*, and OOH* intermediates adsorbed on the V site of the model in **a** with the lowest OER overpotential of 0.25 V. The surface metal atoms are covered by O species. **c** Volcano plot of OER overpotential versus BE difference between OH* and O* species for various sites of $Ni_3Fe_{1-x}V_x$ (oxy)hydroxide models. For each type of reaction site, various structural models are considered, whose detailed information is given in Supplementary Fig. 28 and Supplementary Table 6. The dashed line is a guide for eyes. **d** Left panel: DOS of the model in **a** and the projected DOS on $s$, $p$, and $d$ orbitals. Right panel: projected DOS on the $3d$ orbitals of Ni, Fe, and V atoms in the model. The dashed lines represent the $d$-band center for each element. The Fermi level is shifted to zero. **e** Calculated free-energy diagram of OER on the most active site of $Ni_3Fe_{0.5}V_{0.5}$ (oxy)hydroxide in pH 14 solutions at different potentials ($T = 298$ K). The two-way arrow indicates the overpotential of the rate-limiting step

consider β-NiOOH co-doped by V and Fe atoms with the experimentally optimized doping concentration of Ni:Fe:V = 6:1:1, as well as the systems doped by only V or Fe atom with Ni: V (Fe) = 3:1 (Fig. 7a and Supplementary Fig. 26). The model surfaces are covered by either water molecules or oxygen species that are possibly present in the reaction media. These models with different covered species give very similar results on the catalytic properties (Fig. 7c and Supplementary Table 6).

In the optimized models, the bond lengths between metals and oxygen intermediates are 1.60–1.84, 1.63–1.95, and 1.77–2.05 Å for V, Fe, and Ni, respectively, which are in good agreement with the trend of experimental XAS results. The distinct bond length between O atom and V, Fe, or Ni element is a reflection of their different bond order and bond strength, which is fundamentally governed by the electronic band structure of the material. As revealed by the density of states (DOS) in Fig. 7d, the V, Fe, and Ni atoms in the co-doped $Ni(OH)_2$ have the $d$-band center of

0.09, −2.55, and −2.78 eV, respectively. On the basis of the $d$-band theory[54], the V atoms with higher $d$-band center possess less occupancy of the antibonding states with adsorbed oxygen intermediates, and thus exhibit optimal binding with regard to Ni and Fe atoms (Fig. 7c).

Mulliken charge analysis[55] shows partial charges of ~1.6, 1.0, and $0.8e$ on the V, Fe, and Ni sites, respectively, signifying the stronger metallicity of V atoms and higher chemical activity. Although DFT calculations cannot identify the exact valence for each metal in multi-metal materials, the trend of the partial charges on the V, Fe, and Ni sites obtained from Mulliken charge analysis is consistent with that of the valences of $V^{4+/5+}$, $Fe^{3+}$, and $Ni^{2+}$ estimated on the basis of XANES and XPS. Moreover, the Ni (oxy)hydroxide systems without V doping are half-metal (Supplementary Fig. 27), while the V doping induces finite DOS for the spin-down states near the Fermi level (Fig. 7d), which may help improve the electrical conductivity of the material.

In the previously reported mechanism for $3d$ metal-based (oxy) hydroxide catalysts in alkaline media, the OER undergoes through following four elementary steps[56,57]:

$$^* + OH^- \rightarrow OH^* + e^- \tag{1}$$

$$OH^* + OH^- \rightarrow O^* + H_2O(l) + e^- \tag{2}$$

$$O^* + OH^- \rightarrow OOH^* + e^- \tag{3}$$

$$OOH^* + OH^- \rightarrow^* + O_2(g) + H_2O(l) + e^- \tag{4}$$

where * represents an active site on the catalyst surface; OH*, O*, and OOH* are the oxygen intermediates. To evaluate the OER activity of the Fe and/or V doped or pure Ni (oxy)hydroxide systems, we computed the BEs of oxygen intermediates on various metal sites. The Gibbs free energy for each reaction step and theoretical OER overpotentials were calculated with the standard hydrogen electrode (SHE) method[58].

As displayed by Fig. 7c, the OER overpotentials of doped Ni (oxy)hydroxides follow a volcano-shape relation with the BE difference between OH* and O* (or OOH*) species[59]. In particular, oxygen binding on the Fe and Ni sites is relatively weak, i.e. $E_{OH^*} > 1.15$ eV and $E_{O^*} - E_{OH^*} > 2.24$ eV. As a consequence, formation of OH* and O* species encounters large potential barriers (Supplementary Fig. 29a, b) and will limit the reaction rate of OER process. Large overpotentials of 0.72–0.79 and 0.84–1.08 V are obtained for the Fe and Ni sites, respectively, indicating their low activity for OER. By contrast, the V sites provide much stronger but moderate oxygen binding strength ($E_{OH^*} = 0.47$–0.79 eV and $E_{O^*} - E_{OH^*} = 1.61$–2.23 eV) and give near-optimal BEs of OER intermediates. Reactions to the formation of OH* and O* species are readily accessible, while formation of OOH* experiences the largest potential barrier and limits the OER rate, giving overpotentials of 0.25–0.63 V, which are much lower than those for Ni and Fe sites. Therefore, the highest activity is predicted on the V site of the co-doped Ni (oxy) hydroxide with some of the V and Fe atoms aggregated (Fig. 7b). Such V sites provide strong OOH* binding relative to O* species, and the overpotential is even lower than that of the benchmark catalyst RuO₂, i.e., 0.40 V for the (110) surface according to our calculations. The origin of overpotentials is clearly revealed by the free-energy diagrams as shown in Fig. 7e and Supplementary Fig. 29. The largest potential step at the equilibrium potential ($U = 0.402$ V) indicates the rate-limiting step (RLS) and corresponding overpotential, by overcoming which all the OER steps become downhill and thus can occur spontaneously from the thermodynamic point of view (Supplementary Note 4). In general, the theoretical OER overpotentials follow the same trend as the experimental values: Fe/V co-doped < V-doped < Fe-doped < undoped Ni (oxy)hydroxide (Supplementary Fig. 28 and Supplementary Table 6). DFT calculations show that the OER activity of V sites doped in $Ni_3Fe_{0.5}V_{0.5}$ was greatly enhanced by the surrounding Ni/Fe next-nearest neighbors, and more importantly, the ($Ni_3Fe_{0.5}V_{0.5}$)-OOH models with some of Fe and V atoms aggregated in NiOOH lattices have lower formation energy and higher OER activity than the models with isolated Fe and V atoms. This inference is agree with the statement made by Bell and Calle-Vallejo et al. that for Fe-doped Ni (oxy)hydroxides the surrounding Ni neighbors increase the activity of Fe sites[10,60,61].

## Discussion

In summary, comparative studies on a series of binary and ternary OER catalysts of $Ni_3Fe_{1-x}V_x$ ($0 \leq x \leq 1$) demonstrate that synergistically modulating electronic structure of Ni(OH)₂ by co-doping of Fe and V with optimal doping levels could boost the OER activity of Ni (oxy)hydroxides in alkaline solutions. Notably, $Ni_3Fe_{0.5}V_{0.5}$ features an apparently smaller charge transfer resistance and displays considerably higher specific activity compared to $Ni_3V$ and $Ni_3Fe$, which implies a concerted effect of Fe and V on the OER performance of Ni-based (oxy)hydroxides. The FT and WT analyses of EXAFS data attest the substitution of Fe and V atoms for the Ni sites in Ni(OH)₂ lattices, which is supported by the results obtained from theoretical calculations. The comparative studies of Fe/V co-doped, Fe- or V-doped, and pure Ni (oxy)hydroxides by XPS and soft XAS reveal the synergistic interaction among Fe, V, and Ni cations, rooted from quite different valence electronic configurations of these $3d$ metals. Such interaction subtly influences the electronic structures and local coordination environments of the metals in the ternary catalyst. Accordingly, the XAS results unveil the highly distorted local coordination structure of V and short V–O bond length in $Ni_3Fe_{0.5}V_{0.5}$, which is further shortened under OER conditions. The notable changes in the electronic and geometric structures of V observed in XPS and XAS are echoed by DFT + U calculations, which indicate that the V site has the lowest theoretical overpotential for OER compared with the Ni and Fe sites in $Ni_3Fe_{0.5}V_{0.5}$. Co-doping of Fe and V into Ni(OH)₂ lattices results not only in better metallicity of the material relative to that of solely Fe- or V-doped Ni (oxy)hydroxide, but also in the near-optimal BEs of oxygen intermediates. More importantly, the theoretical calculations indicate that the Fe neighbors near to the V play a crucial role in the enhancement of catalytic activity of the V sites in $Ni_3Fe_{0.5}V_{0.5}$. These findings provide atomic-level insight into the origin of evident enhancement of OER activity of $Ni_3Fe_{0.5}V_{0.5}$. On the basis of the in-depth understanding of the intrinsic relation between electronic structure and OER performance of Ni-based ternary metal (oxy)hydroxide catalysts, it can be envisaged that by using co-doped metals other than Fe, such as Cr, Mn, and Co with different atomic radius, electronegativity, and $d$-band center from those of Fe, the modulation for the local coordination environment and electronic structure of V in Ni (OH)₂ lattices could be regulated, which may further improve the catalytic activity of the Ni/M/V trimetallic catalysts and expand the scope of highly-active Ni(OH)₂-based OER electrocatalysts.

## Methods

**Hydrophilic pretreatment of CFP.** Both sides of the cut-out CFP (thickness of 0.18 mm) were first activated by oxygen plasma treatment with RF frequency of 40 kHz for 3 min (Diener Electronic Plasma-Surface-Technology, Germany), to make the CFP substrate have good hydrophilicity. Subsequently, the pretreated CFP was cleaned by sonication in concentrated nitric acid, deionized water, isopropanol, and acetone for 20 min, respectively, and then kept at 45 °C in a vacuum drier for 5 h.

**Fabrication of $Ni_3Fe_{1-x}V_x$ NS arrays on CFP.** Fe/V co-doped Ni (oxy)hydroxide NS array on CFP was prepared by a hydrothermal method. In a typical fabrication process of $Ni_3Fe_{0.5}V_{0.5}$/CFP, the solution of NiCl₂·6H₂O (0.6 mmol, 142.62 mg), FeCl₃·6H₂O (0.1 mmol, 27.03 mg), and VCl₃ (0.1 mmol, 15.73 mg) in deionized water (40 mL) was magnetically stirred for 10 min to form a homogenous solution, to which urea (4 mmol, 240.24 mg) was added with subsequent stirring for 10 min. Afterwards, the prepared solution was transferred to a 50 mL stainless-steel Teflon-lined autoclave and a piece of the pretreated hydrophilic CFP (3 × 4 cm) was placed upright in the middle of autoclave. Next, the autoclave was sealed and heated in an electric oven at 120 °C for 12 h. After cooling the system to room temperature naturally, the resulting CFP with $Ni_3Fe_{0.5}V_{0.5}$ (oxy)hydroxide NS array was washed with deionized water and ethanol by the assistance of ultrasonication for three times to remove the loosely attached materials, and then dried in vacuum oven at 50 °C overnight. A series of reference electrodes, Ni(OH)₂, Ni₃Fe, Ni₃V, $Ni_3Fe_{0.33}V_{0.67}$, and $Ni_3Fe_{0.67}V_{0.33}$ (oxy)hydroxide NS arrays on CFP, were prepared by the essentially identical procedure. The doping level of Fe and V atoms in the host structure of Ni (oxy)hydroxide was controlled by precisely regulating the molar ratio of Ni/Fe/V salts in the precursor solution, while with the same total amount of metal ions in the initial solutions ($Ni^{2+} + Fe^{3+} + V^{3+} = 0.8$ mmol). For

each hydroxide catalyst, at least three electrodes were prepared and used for the spectroscopic and catalytic measurements.

**Physical and chemical characterizations**. SEM images, EDX, and elemental mappings were measured on a Hitachi SU8220 cold field-emission scanning electron microscope operated at an acceleration voltage of 5 and 15 kV, respectively. BF-TEM and HRTEM were collected on a FEI Tecnai G2 F30 S-TWIN transmission electron microscope with an acceleration voltage of 300 kV. Aberration-corrected HAADF-STEM images, EDX elemental mappings and linear scanning analysis were collected on JEOL ARM200F microscope with STEM aberration corrector operated at 200 kV. XP spectra were taken on a ThermoFisher ESCALAB$^{TM}$ 250Xi surface analysis system using a monochromatized Al Kα small-spot source, and the corresponding BEs were calibrated by referencing the C 1s to 284.8 eV. PXRD patterns were obtained with a Rigaku SmartLab 9.0 using Cu Kα radiation ($\lambda = 1.54056$ Å), and the data were collected in Bragg–Brettano mode in the 2θ range from 10° to 70° at a scan rate of 5° min$^{-1}$. The loading amounts and elemental compositions of catalysts were determined by ICP-OES on an Optima 2000 DV spectrometer (Perkin-Elmer). The as-prepared bi- or trimetallic (oxy) hydroxide array on CFP was immersed in aqua regia for 10 h to completely dissolve the catalyst, and the solution was diluted to 20 mL by deionized water and sonicated for 15 min. All reported ICP-OES results were the average values of at least three independent experiments.

**In-situ EC-Raman measurements**. In-situ EC-Raman spectra were recorded with an XploRA confocal microprobe Raman system. A 50× magnification long working distance (8 mm) objective was used. The wavelength of excitation laser was 785 nm from a He–Ne laser (power was about 4 mW). Raman frequencies were calibrated using Si wafer. The Raman spectra shown in the experiment were collected during 30 s for one single spectrum curve one time, accumulation four times. A custom-made PEC cell with a GCE covered with $Ni_3Fe_{0.5}V_{0.5}$ (oxy)hydroxide catalyst film, a platinum wire counter electrode, and a saturated calomel reference electrode (SCE, 0.242 V versus SHE) was used for EC-Raman measurements. The electrolyte solution (1 M KOH) was saturated with Ar gas before injected into the cell.

**Ex-situ soft and hard XAS measurements**. The soft XAS of Ni $L_{2,3}$-edge, Fe $L_{2,3}$-edge, and V $L_{2,3}$-edge were measured on beamline B12b at the National Synchrotron Radiation Laboratory (NSRL, China) in the total electron yield (TEY) mode by collecting the sample drain current under a vacuum better than $1 \times 10^{-7}$ Pa. The beam from the bending magnet was monochromatized by utilizing a varied line-spacing plane grating and refocused by a toroidal mirror. The energy range is 100–1000 eV with an energy resolution of ~0.2 eV. To optimize the XAS measurements, we collected several XAS spectra at different positions on each sample. No big difference was found among these XAS spectra due to the uniformity of the sample. For annihilating the effect of different sample concentration and measurement conditions on the intensity of characteristic XAS peaks, the data at Ni, Fe, and V L-edges were normalized following the method proposed in literature[62]. The Ni, Fe, and V K-edge XANES and EXAFS spectra were performed on beamline BL14W1 at Shanghai Synchrotron Radiation Facility (SSRF, China) with a ring electron current of 250 mA at 3.5 GeV. The Ni, Fe, and V K-edge XAS spectra of $Ni_3Fe_{1-x}V_x$ (oxy)hydroxide materials were performed in the fluorescence mode using a Lytle detector, while the reference samples ($V_2O_3$, $VO_2$, $V_2O_5$, FeO, and $Fe_2O_3$) with appropriate absorption edge jump were measured in transmission mode. In these conventional fluorescence detection measurements, the background from elastic and Compton scattering was reduced using a combination of Z-1 filters (three absorption lengths of Ti (Mn, Co) for V (Fe, Ni) K-edge spectra) with Soller slits.

**In-situ XAS measurements**. The in-situ XANES and EXAFS data were obtained on beamline BL14W1 at SSRF in the fluorescence mode using a Lytle detector with a step-size of 0.25 eV at room temperature. For the in-situ XAS measurements, an electrochemical workstation (CHI 660E) and a custom-made PEC cell were used. The PEC cell was equipped with a copper frame induced working electrode, a platinum plate counter electrode, and a Hg/HgO (1 M KOH) reference electrode in 1 M KOH (Supplementary Fig. 24). For installation of in-situ XAS setup, the side of $Ni_3Fe_{0.5}V_{0.5}$/CFP electrode covered with a layer of Kapton film was faced to the incident X-rays, while the other side of the CFP covered with catalyst was put in contact with the electrolyte, and the edges of the CFP were fixed to the copper frame electrode with a close contact. Next, the interface was immobilized by a layer of Kapton film, and the inner part of Kapton film was carefully pushed toward the bare side of CFP as close as possible, so as to minimize the influence of electrolyte and bubbles to the acquisition of X-ray signal. Finally, the surface of Kapton film was encapsulated by a flat tool with a coaxial elliptical hole with the assistance of four screws to prevent the electrolyte leakage. During the experiments, the different potentials of 1.15, 1.45, 1.65, and 1.75 V versus RHE were applied to the system.

**Electrochemical measurements**. All electrochemical measurements were carried out at 25 °C on a CHI 660E potentiostat. A three-electrode H-shape cell was used with the as-prepared Ni-based (oxy)hydroxide/CFP (0.2 cm$^2$) as the working electrode, a platinum mesh (Tjaida) as the counter electrode, and a Hg/HgO (1 M

KOH, Tjaida) as the reference electrode. Prior to each electrochemical experiment, the cell was washed and stored in 0.5 M $H_2SO_4$; the counter electrode was cleaned in aqua regia for 30 s to remove any oxidative and deposited species during OER process; the electrolyte (1 M KOH) was degassed by bubbling oxygen for 30 min; the reference electrode was corrected against another unused Hg/HgO electrode stored in 1 M KOH solution. The measured potentials versus Hg/HgO were converted to the potentials versus RHE by the following equation:

$$E_{RHE} = E_{Hg/HgO} + 0.059\,pH + E^o_{Hg/HgO} (E^o_{Hg/HgO} = 0.098V\ versus\ SHE) \quad (5)$$

Five cycles of CV were executed at a scan rate of 50 mV s$^{-1}$ prior to the measurement of OER polarization curves at 5 mV s$^{-1}$, and the Tafel slopes were derived from the corresponding OER polarization curves. For all polarization curves presented in the paper, the $iR$ values were manually corrected with the series resistance ($R_s$) on the basis of the equation:

$$E_{RHE} = E_{Hg/HgO} + 0.059\,pH + E^o_{Hg/HgO} - iR_s \quad (6)$$

The compensated ohmic $R_s$ values were obtained from the fittings of electrochemical impedance spectra.

**Computational method**. Spin-polarized DFT calculations were performed by the Vienna ab initio simulation package (VASP), using the planewave basis with energy cutoff of 500 eV (ref. [63]), the projector augmented wave (PAW) potentials[64], and the PBE functional for the exchange-correlation energy[65]. Grimme's semi-empirical DFT-D3 scheme of dispersion correction was adopted to describe the van der Waals (vdW) interactions in layered materials[66]. The Hubbard-U correction was applied for better description of the localized d-electrons of Ni, Fe, and V in their (oxy)hydroxides[67]. We chose an effective $U-J$ value of 3.0 eV for V and Fe and 5.5 eV for Ni atoms, close to the literature values[56,68,69]. Mulliken charge analysis[55] was performed by CASTEP code[70] using the planewave basis with an energy cutoff of 1000 eV and norm-conserving pseudopotentials.

## Data availability
The data that support the findings of this study are available from the corresponding authors upon reasonable request.

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

## Acknowledgements

The authors acknowledge financial support from the Natural Science Foundation of China (Grant Nos. 21673028, 11435012, and 21373040) and the Basic Research Program of China (Grant No. 2014CB239402).

## Author contributions

J.J. conceived the project, performed most of the experimental work, and drafted part of the manuscript. M.W. conceived and supervised the project, wrote the main part of the paper, and revised the entire paper. J.J., F.S., H.Z., Z.J., W.H., and W.Y. conducted the XAS experiments and analyzed the data. W.Y. wrote the part on XAS results. S.Z. and J.Z. performed the DFT calculations, and S.Z. wrote the part on DFT calculations. J.D. and J. L. made the EC-Raman experiments.

## Additional information

**Competing interests:** The authors declare no competing interests.



