## [Peer Review File · Nature Communications]

Reviewers' comments:

Reviewer #1 (Remarks to the Author):

Jiang et al. report doping of V into NiFe-LDH to enhance its activity for the oxygen evolution reaction. Both activity and stability of the composite electrodes are impressive. NiFe hydroxides or LDH are a hot topic due to their high activity for OER in alkaline media. The authors' work shows that V-doping can further enhance the activity without compromising the stability of the material (and composite electrode). As such, the work should gain the attention of the electrocatalysis and solar fuels communities.

The oxide is characterized by ex situ XRD, TEM, EDX, XPS, XAS as well as in situ XAS and Raman spectroscopy. The in situ experiments show that the active state is different from the starting material, namely (Ni,Fe,V)OOH. The experimental measurements are complemented by DFT calculations of the active surface to gain insight into the mechanism. It is clearly demonstrated that the intrinsic activity increases with V doping. Unfortunately, this strategy is neither new (ref. 27) nor is the novelty of the "synergistic effect" of V clearly discussed and it is unclear how exactly other electrocatalysts may benefit from it. Moreover, I disagree with the assignment of the vanadium species and details of the mechanism.

Overall, the topic and comprehensiveness of the results may be suitable for publication in Nature Communications after its weak points and further comments have been addressed:

1) Novelty. The influence of Fe/V co-doping was already studied by Singh & Singh for a spinel (ref. 27). How does the authors' work go beyond ref. 27? What are the new insights?

2) Benefit of V doping.

a) Please state clearly under which conditions and in which materials V is reduced (or oxidized?) relative to a meaningful standard. The expression "charge transfer" is too ambiguous (see below). This would help to follow changes of the electronic structure and quantify them.

b) P20. "[V atoms] exhibit enhanced oxygen binding and OER activity". An optimal binding strength is required for catalysis (i.e. Sabatier's principle) as also seen in Fig. 7c. Perhaps "optimal binding" is clearer for the readers and please refer to Fig. 7c.

c) What exactly distinguishes V from Fe and Ni? The discussion suggests that Fe and Ni donate electrons to V thus reducing it. It is unclear how electrons on V are beneficial for OER where holes need to be transferred to oxygen.

d) P21. "[These findings] give some helpful hints for reasonably designing new V-containing catalysts" What are these hints specifically? Are they generalizable to other oxides? Are they new?

3) I disagree with the assignment of the vanadium species.

a) The short V-O bond lengths are much shorter than previously reported for V⁴⁺ oxides and better match that of a V⁵⁺ oxide. The high pre-edge in the V-K XANES also supports V⁵⁺, which was also found by XPS. I note that V⁵⁺ in V₂O₅ is highly distorted. See also the discussion in http://www.exafsmaterials.com/Literature/12_V_XANES_1984.pdf
The bond length would not be "ultra-short" for V⁵⁺. Please add additional discussion about the formal V valence.

b) Does the electronic interaction between Fe, Ni and V affect the formal valence obtainable by XAS or does it only manifest in the M-O hybridization?

c) The EXAFS at the V-K edge does not support doping of V into the FeNi oxide lattice as there is only a single FT peak above the noise level. Is there additional evidence that V occupies a Fe or Ni position in the lattice? Could V be interstitial or on top of the LDH layers? Both would straightforwardly explain

the lack of peaks at high R in the EXAFS FT.

d) Please show the "NiFeV" spectra at the Ni-K, Fe-K and V-K edge in a single plot to support the short V-O distance for this review. If this figure is included in revised manuscript, state clearly that phase contributions of Ni, Fe, V are not identical during FT (i.e. the reduced distances on the x-axis are not identical).

e) The V-K pre-edge is lowered after OER. This could indicate a change in symmetry. The V-L spectra cannot exclude a change in symmetry without detailed analysis of the multiplet structure.

f) P16. "the V-O distance is also shrunk from 1.69 to 1.66 Å" The V-O bond is not shorter within the uncertainty given in the supporting table.

3) How many electrodes were prepared for each oxide? Are the measurements reproducible?

4) Mechanism

a) I like to point out that the mechanism of the LDH is still actively disused, e.g. in onlinelibrary.wiley.com/doi/10.1002/aenm.201600621/full

Can the author's data support one of the previous proposals or the selection of the used reaction steps?

b) The point of zero charge (PZC) of oxides is below pH 12, so at pH 14, the equilibrium should be shifted toward *O. I disagree that it should be a possible RLS since the solution thermodynamics favor its generation. The potential of charge neutrality is also much lower than the onset of the OER on oxides.

Concept of PZC: <http://old.iupac.org/publications/pac/1976/pdf/4804x0415.pdf> and [http://www.gly.uga.edu/railsback/Fundamentals/8150PointofZeroChar ge05Pt1P.pdf](http://www.gly.uga.edu/railsback/Fundamentals/8150PointofZeroCharge05Pt1P.pdf)

5) EXAFS analysis

a) Please mention for the readers that Fig. 3 and 6 show the reduced distance. The interatomic distance is determined by the fits.

b) Please state the details of EXAFS FT. What was the windowing function and its parameters? What K-space range was used? What R-space range was analyzed/fit? This information is necessary for reproduction of the results

c) Phases from experimental references were used. Which reference materials were used? How were they extracted. This can significantly affect the obtained fit values. Could the low V-O distance be an artefact of the experimental phase function?

d) How is the uncertainty in the tables of the EXAFS analysis calculated? Is it the parameter error?

e) How do the fit results compare to ref. 10 and 14? This should also be included in the revision for the readers.

6) The intensity (or better area under the curve) of soft XAS spectra does depend on the number of holes but also on other factors

a) How were the spectra normalized?

b) How do the soft XAS spectra look at different positions on the sample? Are the changes significant?

c) Can the observed changes due to change in the total absorption when another element is added to the oxide?

d) A significant change in holes on the element should also lead to a shift of the peak position, which is only observed for V.

7) DFT

a) Which metal valences were found in the DFT calculations? Do they match the trends obtained from XAS? The DFT and XAS results could be better connected.

- b) Why are there different circles of the same color in Fig. 7c? What models do they belong to? Please provide a supporting table with detail or other means of documentation
- c) Are surrounding atoms and the electrolyte considered in calculations of the oxygen adsorption? Please add detail in the main text
- d) P.19. "The highest activity is achieved". This refers to a calculation and thus "is predicted".

8) Errors or ambiguous discussion

- a) P10 "For Ni₃Fe (oxy)hydroxide, the unpaired electron in the n-symmetry (t_{2g}) d-orbital of Fe³⁺". The Fe-L spectrum in Fig. 4e is clearly that of high spin Fe³⁺ (t_{2g}³ e_g²). Thus, there are three unpaired t_{2g} electrons. I also recommend giving the t_{2g} and e_g occupancies instead of the d-electron count on P10, which would make it easier to follow. Lastly, the soft XAS spectra should be discussed in terms of t_{2g}/e_g occupancy or at least whether the ion is high or low spin, which can be done, e.g., using the branching ratio (L₃/L₂+L₃).
- b) The authors use "charge transfer" frequently. It is confusing and imprecise. Which charge carrier? Electron or hole? It would be clearer to state which element would be reduced or oxidized.
- c) P14. The ECSA was not measured. It is obtained by dividing the capacitance of the sample by that of a perfectly flat reference of the same material, see ref. 3. The used normalization corrects for difference in roughness among the samples and is thus useful but should not be called ECSA and cannot be compared to catalysts on other substrates.
- d) Supplemental Figures S5+S6 do not show "atomic resolution". They are blurry and individual atomic columns are not resolved.
- e) Supplemental P37. NiOOH was not found by PXRD or HRTEM. It is only detected by Raman and in situ XAS.

9) Minor/typo:

- a) Supplemental Fig. 10. What does the label Ni stand for? Ni metal? Ni(OH)₂? Please add simple Ni oxide references.
- b) The "Janus face property" is called amphoteric in chemistry or is something else meant?
- c) P14. Which of the two charge transfer resistances is reduced?
- d) Supplemental Table 9. Please add potential at which EIS was performed and refer to circuit
- d) Please index PXRD patterns. Which reflections change/shift? Are those assigned to the metal oxide slabs or to the distance between the metal oxide slabs (of the LDH)?
- e) I was confused by Fig. 1, where the substrate looks like a metal mesh. The carbon fiber substrate is not as ordered as shown there.

Reviewer #2 (Remarks to the Author):

This is a very comprehensive paper of very high scientific quality and considerable topical interest. The development of metal oxide electrocatalytic materials which catalyse the electro-oxidation of water to generate molecular oxygen is a grand challenge in energy science. Doping metal oxide lattices with small quantities of foreign metal atoms has been shown to be a useful partway to lower the overpotential required to drive the water oxidation at a fixed current density (say 10 - 20 mA/cm²). The process of doping nickel oxyhydroxide with iron has been shown over the last few years, to significantly enhance the catalytic activity of the bimetallic catalyst.

In the present paper the authors have convincingly shown that the addition of vanadium to the bimetallic catalyst can result in even better catalytic activity. An overpotential of ca. 200 mV at 10 mA/cm² with a Tafel slope of ca. 39 mV/dech has been measured for a

Ni₃Fe_{0.5}V_{0.5} oxyhydroxide material. These are impressive performance indicators. This is a very significant result.

The paper is very complete. The characterization and analysis of the material and its performance under operating conditions has been rigorously examined. Furthermore the paper contains significant DFT analysis as an aid to mechanistic interpretation.

Given the quality of the science and its inherent broad interest and topicality, I strongly recommend that this paper be accepted for publication.

Reviewer #3 (Remarks to the Author):

This manuscript aims to unveil OER improvements through Fe/V co-doping in Ni(OH)₂ by comprehensive spectroscopic studies including XAFS, sXAS and Raman along with advanced TEM of catalysts ex-situ and in-situ of OER. The experimental results have been combined with calculation to get a conclusion that V site is the active sites in this catalysts. the methodology is appealing and should deliver new scientific insights for this important catalyst. unfortunately the current version doesn't get there. therefore I recommend a major revision plus another round review. in the revision the author needs to carefully address those questions:

- 1) V in situ XAFS needs to be conducted since this is the proposed active sites. I understand that in situ work for V is a bit challenge but it is still doable.
- 2) from in situ XAFS, the structure of Ni/Fe in the active catalyst under OER shall be resolved; therefore this structure information shall be used in calculation rather than using models from literature.
- 3) the manuscript only "proves" the knowing knowledge that V is the active sites from calculation (optimal binding energy for OER intermediates at V site) and XAFS (ultra short V-O bond). a more deep discussion that how such local structural change led to the calculated results are need and this probably unveil atomic level new insights of the high OER catalysis in this known catalyst.
- 4) in addition to those above main concerns the claim of a uniform substitution of Fe/V into Ni(OH)₂ lattice to substitute Ni sites cannot be supported by TEM-EDX mapping. EXAFS at V edge also seems against the claim.

Title: Atomic-level insight into super-efficient electrocatalytic oxygen evolution on iron and vanadium co-doped nickel (oxy)hydroxide

Dear editor,

Thank you for your e-mail of Feb. 17. We appreciate referees' professional comments. The manuscript has been carefully revised according to the referees' comments. The changes in the manuscript and the responses to referees' questions are listed below. Some new experimental results have been added to the revised manuscript. We would be grateful if you and referees find the revised manuscript acceptable for publication in *Nature Communications*.

Responses to the comments of Referee #1:

Comments: *“Jiang et al. report doping of V into NiFe-LDH to enhance its activity for the oxygen evolution reaction. Both activity and stability of the composite electrodes are impressive. NiFe hydroxides or LDH are a hot topic due to their high activity for OER in alkaline media. The authors' work shows that V-doping can further enhance the activity without compromising the stability of the material (and composite electrode). As such, the work should gain the attention of the electrocatalysis and solar fuels communities.*

The oxide is characterized by ex situ XRD, TEM, EDX, XPS, XAS as well as in situ XAS and Raman spectroscopy. The in situ experiments show that the active state is different from the starting material, namely (Ni,Fe,V)OOH. The experimental measurements are complemented by DFT calculations of the active surface to gain insight into the mechanism. It is clearly demonstrated that the intrinsic activity increases with V doping. Unfortunately, this strategy is neither new (ref. 27) nor is the novelty of the “synergistic effect” of V clearly discussed and it is unclear how exactly other electrocatalysts may benefit from it. Moreover, I disagree with the assignment of the vanadium species and details of the mechanism.

Overall, the topic and comprehensiveness of the results may be suitable for publication in Nature Communications after its weak points and further comments have been addressed.”

Question 1: *“Novelty. The influence of Fe/V co-doping was already studied by Singh &*

Singh for a spinel (ref. 27). How does the authors' work go beyond ref. 27? What are the new insights?"

Response 1: The most important difference of our work from the work reported by Singh & Singh [ref. 27, *Int. J. Hydrogen Energy* **35**, 3243–3248 (2010)] is that in our work *Fe and V were co-doped into the Ni(OH)₂ lattices, whereas in ref. 27 vanadium was doped into the NiFe₂O₄ spinel lattices* as evidenced by the XRD patterns. The XRD patterns of the Ni₃Fe_{1-x}V_x (0 ≤ x ≤ 1) (oxy)hydroxide catalysts prepared in the present work match well with that of hexagonal α-Ni(OH)₂ (JCPDS Card No. 38-0715), and the diffraction peaks for the cubic NiFe₂O₄ spinel (JCPDS Card No. 23-1119) are not observed. Notably, the doping atoms of Fe and V substitute for the positions of Ni in our as-prepared catalysts, while the V atoms are just incorporated into the Fe, not Ni, sites in ref. 27; and moreover, all metal ions are located in an octahedral coordination geometry in our Ni₃Fe_{0.5}V_{0.5} catalyst, while the Ni, Fe and V ions have different coordination geometries, namely tetrahedral for Ni and octahedral for Fe and V, in the catalyst reported in ref. 27.

In addition, the NiFe_{2-x}V_xO₄ catalysts were only characterized by IR and XRD in ref. 27, which did not provide convincing evidence for substitutionally doping of V into the lattices of host materials, let alone the information about the active sites of the multimetal catalysts under OER conditions. By contrast, we have employed many advanced techniques, such as *ex-situ* soft and *in-situ* hard XAS, *in-situ* EC-Raman spectroscopy, SEM, HR-TEM and atomic-resolution BF-TEM, aberration-corrected HAADF-STEM, EDX, XPS, and PXRD, as well as computational studies, to provide convincing evidence for co-doping of V and Fe into the Ni(OH)₂ lattices and to reveal how V dopant interplays with other metal ions co-existing in the trimetallic catalyst, and how the doped V cations contribute to the high OER activity of the host material. As we have stated in the Introduction, to date, there is no report on in-depth spectroscopic studies of local coordination environment and electronic structure for the V-containing bi- and trimetal (oxy)hydroxide OER catalysts in both rest and activated states. This is the key point and the new insight of the present work.

It is also worthy of note that we fabricated the Ni₃Fe_{1-x}V_x electrodes with a protocol completely different from that used in ref. 27, in which the NiFe_{2-x}V_xO₄ electrodes were fabricated by multi-steps: (i) the precursor of NiFe_{2-x}V_xO₄ was precipitated from the basic solution (pH 11) at 70 °C; (ii) the solid powder obtained was dried at 100 °C for 24 h and then treated at 600 °C for 5 h; (iii) the as-prepared NiFe_{2-x}V_xO₄ catalyst was mixed with glycerol to form a slurry, which was painted onto a Ni substrate; finally

(iv) the $\text{NiFe}_{2-x}\text{V}_x\text{O}_4$ film electrode was treated at 350 °C for 1.5 h. By contrast, we directly grew $\text{Ni}_3\text{Fe}_{1-x}\text{V}_x$ (oxy)hydroxides on the carbon fiber paper (CFP) by hydrothermal synthesis at 120 °C, without post high temperature treatment. Compared to the literature method, the advantages of the hydrothermal synthesis protocol are (i) more operation-convenient and less energy-consuming, (ii) forming better contact between the OER catalyst and the conductive substrate, and (iii) resulting in controllable morphology of the catalyst (an ultrathin nanosheet array on CFP) to have more active sites exposed to electrolyte. As a result, our optimized electrode of the $\text{Ni}_3\text{Fe}_{0.5}\text{V}_{0.5}$ (oxy)hydroxide displays much higher activity and better stability for OER than the electrode with the $\text{NiFe}_{2-x}\text{V}_x\text{O}_4$ reported in ref. 27. Specifically, the overpotential at 122 mA cm^{-2} current density is 269 mV for our $\text{Ni}_3\text{Fe}_{0.5}\text{V}_{0.5}$ (oxy)hydroxide electrode versus 394 mV for the $\text{NiFe}_{2-x}\text{V}_x\text{O}_4$ electrode in 1 M KOH solution reported in ref. 27. Moreover, the $\text{Ni}_3\text{Fe}_{0.5}\text{V}_{0.5}/\text{CFP}$ electrode displays excellent stability under testing conditions, even at the current density of 100 mA cm^{-2} over 60 h of electrolysis.

Question 2: *“Benefit of V doping.”*

Question 2a: *“Please state clearly under which conditions and in which materials V is reduced (or oxidized?) relative to a meaningful standard. The expression “charge transfer” is too ambiguous (see below). This would help to follow changes of the electronic structure and quantify them.”*

Response 2a: Thanks for your good advice. We try to make a clear statement for this question by adding the following sentence to the manuscript at page 11: “The comparative analyses of XPS and XANES spectra suggest that co-doping of Fe together with V into $\text{Ni}(\text{OH})_2$ lattices results in more electron transfer to the V in $\text{Ni}_3\text{Fe}_{0.5}\text{V}_{0.5}$ compared to that in Ni_3V (Fig. 4c,f and Supplementary Fig. 16c).”

The expression “charge transfer” has been revised to “partial electron transfer” in the entire manuscript.

Question 2b: *“P20. “[V atoms] exhibit enhanced oxygen binding and OER activity”. An optimal binding strength is required for catalysis (i.e. Sabatier’s principle) as also seen in Fig. 7c. Perhaps “optimal binding” is clearer for the readers and please refer to Fig. 7c.”*

Response 2b: Thanks for your suggestion. To be clearer, the relevant sentence at page 19 (in the revised manuscript) has been revised to “On the basis of the *d*-band theory⁵⁴,

the V atoms with higher d -band center possess less occupancy of the antibonding states with adsorbed oxygen intermediates, and thus exhibit optimal binding with regard to Ni and Fe atoms (Fig. 7c).”

Question 2c: “*What exactly distinguishes V from Fe and Ni? The discussion suggests that Fe and Ni donate electrons to V thus reducing it. It is unclear how electrons on V are beneficial for OER where holes need to be transferred to oxygen.*”

Response 2c: In our opinion, the low t_{2g} and e_g occupancies of V^{n+} ($n = 3, 4,$ and 5) as compared to those of Fe^{3+} and Ni^{2+} make the redox property and the OER activity of the V sites distinguishing from those of Fe and Ni sites in the $Ni_3Fe_{0.5}V_{0.5}$ (oxy)hydroxide system.

In $Ni_3Fe_{1-x}V_x$ (oxy)hydroxide catalysts, the Fe^{3+} and Ni^{2+} with half-full e_g orbitals would have very weak bonding with adsorbed oxygen species, whereas the V^{4+} and V^{5+} with e_g^0 orbitals would form too strong bonding with adsorbed oxygen species. To get high OER activity, the bonding strength between transition metal and adsorbed oxygen species should be optimized to fulfill the Sabatier principle [*Ber. Deutsch. Chem. Gesellschaft* **44**, 1984–2001 (1911)]. With increase of the electron density on V by partial electron transfer from Fe and Ni to V through the bridging O^{2-} ions, the high valence states of V could be stabilized under OER conditions, and more importantly, the strong bond strength between V and adsorbed oxygen species could be tuned to a moderate bond strength, which would benefit for releasing O_2 from the V site in OER. This part of explanation has been added to page 12.

Question 2d: “*P21. “[These findings] give some helpful hints for reasonably designing new V-containing catalysts” What are these hints specifically? Are they generalizable to other oxides? Are they new?*”

Response 2d: Specifically, the findings of this work give the following hints for reasonably designing new V-containing OER catalysts:

(i) The spectroscopic findings and computational results from this work indicate that compared to the sole V-doped Ni (oxy)hydroxide, the synergetic interaction of the Fe/V dopants and the Ni host in the Ni/Fe/V (oxy)hydroxide leads to a more distorted coordination geometry and a disturbed electronic structure of the doped V, which makes the crucial contribution to the superior OER performance of the Ni/Fe/V trimetallic catalyst.

(ii) On the basis of these findings, it can be envisaged that by using co-doped metals

other than Fe, such as Cr, Mn, and Co with different atomic radius, electronegativity, and *d*-band center from those of Fe, the modulation for the local coordination environment and electronic structure of V in the Ni(OH)₂ lattices could be regulated, which may further improve the OER catalytic activity of the Ni/M/V trimetallic catalysts and expand the scope of highly-active Ni(OH)₂-based OER electrocatalysts.

The sentence, “[These findings] give some helpful hints for reasonably designing new V-containing catalysts” at page 23, has been revised according to the above paragraphs.

Question 3: *“I disagree with the assignment of the vanadium species.”*

Question 3a: *“The short V-O bond lengths are much shorter than previously reported for V⁴⁺ oxides and better match that of a V⁵⁺ oxide. The high pre-edge in the V-K XANES also supports V⁵⁺, which was also found by XPS. I note that V⁵⁺ in V₂O₅ is highly distorted. See also the discussion in http://www.exafsmaterials.com/Literature/12_V_XANES_1984.pdf. The bond length would not be “ultra-short” for V⁵⁺. Please add additional discussion about the formal V valence.”*

Response 3a: Thanks for informing us this valuable reference. It reveals that in V *K*-edge XANES spectra, the intensity of the pre-edge peak depends predominantly on central site symmetry, while the absorption edge position is correlated to the oxidation state of central site. The intensity of the pre-edge peaks enhances in an ascending order of V₂O₃ < V₂O₄ < V₂O₅, predominantly due to the distortion of the coordination geometry from octahedral symmetry. Meanwhile, the absorption edge position of V positively shifts to higher energies with valence increase. In our case, the V *K*-edge XANES spectra of Ni₃V and Ni₃Fe_{0.5}V_{0.5} exhibit intense pre-edge peaks (Supplementary Fig. 15c), indicating the distorted coordination environment around V atoms in these materials. More interestingly, Ni₃Fe_{0.5}V_{0.5} shows a higher pre-edge peak than that of Ni₃V in the V *K*-edge XANES, implying a higher degree of octahedral geometry distortion at the V sites in Ni₃Fe_{0.5}V_{0.5} compared to those in Ni₃V. The *K*-edge absorption positions of Ni₃V and Ni₃Fe_{0.5}V_{0.5} are more close to those of VO₂ and V₂O₅ than to that of V₂O₃ (inset of Supplementary Fig. 15c), suggesting that the majority of V ions are in the formal valences of +4 and +5 in both catalysts. Consistently, the XP spectra also show that the V atoms are predominantly in high oxidation states (+4 and +5) in Ni₃V and Ni₃Fe_{0.5}V_{0.5}, together with a minority of V³⁺ (Fig. 4c). Additionally, the short V–O bond lengths of 1.72 Å for Ni₃Fe_{0.5}V_{0.5} and 1.69

Å for Ni₃V, obtained from the fitting parameters of V *K*-edge EXFAS curves by using two V–O paths, are close to those of the V–O bond lengths reported for V⁴⁺ and V⁵⁺ oxides [*Phys. Rev. B* **30**, 5596–5610 (1984)].

The related original discussion at page 8 has been revised to “Importantly, the V *K*-edge XANES spectra of Ni₃V and Ni₃Fe_{0.5}V_{0.5} exhibit intense pre-edge peaks (Supplementary Fig. 15c), indicating the distorted coordination environment around V atoms in these materials⁴². More interestingly, Ni₃Fe_{0.5}V_{0.5} shows a higher pre-edge peak than that of Ni₃V in the V *K*-edge XANES, implying a higher degree of octahedral geometry distortion at the V sites in Ni₃Fe_{0.5}V_{0.5} compared to those in Ni₃V. Additionally, the *K*-edge absorption positions of Ni₃V and Ni₃Fe_{0.5}V_{0.5} are more close to those of VO₂ and V₂O₅ than to that of V₂O₃ (inset of Supplementary Fig. 15c), suggesting that the majority of V ions are in the formal valences of +4 and +5 in both catalysts.”

In addition, the corresponding original discussion on the V 2*p*_{3/2} peaks at page 9 was also revised as follows: “The V 2*p*_{3/2} peak (Fig. 4c) can be deconvoluted into three peaks located at 516.2 eV (V⁵⁺), 515.1 eV (V⁴⁺), and 514.4 eV (V³⁺) (refs 26,28,29), demonstrating that the V atoms are predominantly in high oxidation states (+4 and +5) in Ni₃V and Ni₃Fe_{0.5}V_{0.5}, together with a minority of V³⁺, which is consistent with the results obtained from V *K*-edge XANES spectra.”

Question 3b: “*Does the electronic interaction between Fe, Ni and V affect the formal valence obtainable by XAS or does it only manifest in the M–O hybridization?*”

Response 3b: Although the partial electron transfer from Ni and Fe to V in Ni₃Fe_{0.5}V_{0.5} (oxy)hydroxide is evidenced by XPS and *L*-edge XAS spectra (Fig. 4), the electronic interaction between Fe, Ni and V in the Ni-based ternary material is not strong enough to alter the formal valence of the metals. The evidence observed from XPS and *L*-edge XAS only manifests the M–O–M' hybridization with partial electron transfer among Ni, Fe and V through the oxygen bridges.

Question 3c: “*The EXAFS at the V-K edge does not support doping of V into the FeNi oxide lattice as there is only a single FT peak above the noise level. Is there additional evidence that V occupies a Fe or Ni position in the lattice? Could V be interstitial or on top of the LDH layers? Both would straightforwardly explain the lack of peaks at high R in the EXAFS FT.*”

Response 3c: These are good questions. Yes, for Ni₃Fe_{0.5}V_{0.5}, the EXAFS FT curve at

the V *K*-edge displays only a prominent V–O peak and the V–M (M = Fe, Ni or V) peak is weaker than that of Ni₃V. This is most likely caused by the large distortion of the local structure of the V substituting the site of Ni in Ni(OH)₂ lattices. The possible reasons and additional evidence for V occupancy of a Ni position in the Ni(OH)₂ lattice are as follows:

(i) The large distortion is also reflected from the V–O peak. Compared with the single V–O path fitting, better match with the experimental data is obtained if two V–O paths are included in fitting the V–O peak. This implies the large local distortion around the substitutional V atoms. As a result, the distance distribution of V–M (M = Fe, Ni or V) coordinations is in a wide range, and the different coordination atoms of Fe, Ni and V also make the static disorder of the V–M pairs very large. Consequently, the V–M (M = Fe, Ni or V) coordination peak in the FT curve is significantly damped.

(ii) To further confirm the V substitution for Ni in the Ni(OH)₂ lattice, the wavelet transform (WT) analysis of the V *K*-edge EXAFS data was made, which reveals a similar feature of Ni₃Fe_{0.5}V_{0.5} to that of Ni₃V, i.e., the WT maximum appears at the cross point of $R = 2.8 \text{ \AA}$ and $k = 7.8 \text{ \AA}^{-1}$ (Supplementary Fig. 7), implying the presence of V–Fe/Ni scatterings at a distance of around 2.8 Å surrounding V atoms. This affords direct evidence for the substitution of V atoms for the Ni sites in the Ni(OH)₂ lattice.

(iii) We also made the calculation of the EXAFS spectra by assuming V adsorption on the LDH layer or occupying the interstitial position. It reveals that in both cases the calculated spectra are quite different from the experimental spectra (Supplementary Fig. 8).

(iv) Furthermore, DFT calculations suggest that V atoms initially placed on the top site of surface Ni or O atoms are relaxed to the interstitial between two LDH layers after structure optimization. The LDH structure with interstitial doping is noticeably buckled, with formation energy of –3.73 eV per V atom, less stable with regard to LDH with substitutional doping (–5.07 eV per V atom) (Supplementary Figs 9 and 10), supporting that V occupies a Ni position in the (oxy)hydroxide lattice rather than the interstitial or top positions of the LDH layers.

(v) It has been revealed by many experimental and calculation studies that transition-metal ions are prone to substitute for the Ni sites in LDH [*J. Am. Chem. Soc.* **139**, 3876–3879 (2018); *Adv. Mater.* **30**, 1706279 (2018); *J. Am. Chem. Soc.* **139**, 2070–2082 (2017); *Nat. Energy* **1**, 16053 (2016); *Nat. Commun.* **7**, 11981 (2016); *Nat. Commun.* **7**, 12324 (2016); *J. Am. Chem. Soc.* **138**, 5603–5614 (2016); *J. Am. Chem. Soc.* **137**, 1305–1313 (2015); *J. Am. Chem. Soc.* **137**, 15090–15093. (2015); *J. Am.*

Chem. Soc. **135**, 8452–8455 (2013)]. Our results are in agreement with these results.

The related additional evidence and discussion on the substitution of V atoms for the Ni sites in the Ni(OH)₂ lattice has been added to pages 6–7 in the revised manuscript, and the additional figures have been added to the Supplementary Information as Supplementary Figs 7, 8, 9, and 10.

Question 3d: “Please show the “NiFeV” spectra at the Ni-K, Fe-K and V-K edge in a single plot to support the short V–O distance for this review. If this figure is included in revised manuscript, state clearly that phase contributions of Ni, Fe, V are not identical during FT (i.e. the reduced distances on the x-axis are not identical).”

Response 3d: According to this suggestion, we display the FT-EXAFS curves of Ni, Fe and V *K*-edge data for Ni₃Fe_{0.5}V_{0.5} in a single plot, which has been added to the Supplementary Information as Supplementary Fig. 11. For *ex-situ* FT-EXAFS, the *k*-ranges of the FT are 3.0–12.4 Å⁻¹ for Ni and Fe *K*-edges and 3.0–12.5 Å⁻¹ for V *K*-edge, and the phase-shifts are not corrected.

Besides, the related discussion at page 7 has been revised to: “On the other side, from Supplementary Fig. 11, the nearest-neighbor FT peak position of V is shifted to the lower-*R* side and the second coordination peak to higher-*R* side with apparently reduced intensity as compared to that of Fe. This implies the remarkable different local environment of the substitutional V from that of Fe in Ni₃Fe_{0.5}V_{0.5}.”

Question 3e: “The V-K pre-edge is lowered after OER. This could indicate a change in symmetry. The V-L spectra cannot exclude a change in symmetry without detailed analysis of the multiplet structure.”

Response 3e: Please notice that a change in symmetry could lead to not only a change of the intensity of the pre-edge peak at V *K*-edge, but also a significant change of the spectral feature of the entire XANES spectrum due to the sensitivity of XANES to the geometric structure around the V atoms, just like what is shown in *Phys. Rev. B* **30**, 5596–5610 (1984). However, after the OER at 1.75 V, except for a decrease in the pre-edge peak intensity (the decrease extent is much smaller than that caused by the change in symmetry), our V *K*-edge XANES shows identical spectral features (including shape and intensity of other characteristic peaks) to those measured before OER (original Fig. 6f). Therefore, we consider that the slight decrease in the intensity of the pre-edge peak at V *K*-edge for Ni₃Fe_{0.5}V_{0.5} is most likely caused by electron transfer, rather than by a change in symmetry. Similarly, no other obvious change is

visible at the V *L*-edge spectra after OER (Supplementary Fig. 25), except for the decrease in the intensity of the characteristic peaks. These observations indicate no change of the coordination environment around V. That is to say, V atoms still occupy the sites of Ni after OER.

Nevertheless, as we have successfully conducted the *in-situ* V *K*-edge XAS experiment of Ni₃Fe_{0.5}V_{0.5} according to the comment of Referee #3, we used the new V *K*-edge XANES spectra to replace the original Fig. 6f, and the related discussion on this problem has been rewritten in the revised manuscript at pages 17–18.

Question 3f: “P16. “the V–O distance is also shrunk from 1.69 to 1.66 Å” The V–O bond is not shorter within the uncertainty given in the supporting table.”

Response 3f: We thank the referee for reminding us of this problem. In the original Supplementary Table 4, the uncertainty of the V–O bond length was 0.04 Å, much larger than the common uncertainty (0.01 Å) of the first nearest neighbor. We also noticed that the fitting quality in the original Fig. 6g is not good when only a single V–O path is considered. Therefore, a single V–O path is inadequate to fit the first peak in the V *K*-edge FT curve of Ni₃Fe_{0.5}V_{0.5}. To solve this problem, we re-fit the data of Ni₃Fe_{0.5}V_{0.5} by using two V–O paths, which yield better matches between the experimental data and fitting curves (see figure below). The refitting also indicates the shrinkage of the V–O distance from 1.72 Å before OER to 1.66 Å after OER, with an uncertainty of the V–O bond length of 0.01 Å (see table below). Nevertheless, based on the results of *in-situ* V *K*-edge XAS experiments of Ni₃Fe_{0.5}V_{0.5}, we used the new V *K*-edge FT-EXAFS spectra fitted by two V–O paths to replace the original Fig. 6g. Besides, the original Fig. 6g has been removed together with the related fitting parameters in Supplementary Table 4, and the fitting data from the new spectra have been added to Supplementary Table 8. The description for EXAFS data analysis has been added to Supplementary Methods.

Revised original Fig. 6g | FT-EXAFS spectrum and fit of V K-edge of $\text{Ni}_3\text{Fe}_{0.5}\text{V}_{0.5}$ before and after OER at 1.75 V electrolysis in 1 M KOH. The circle represents the fitting values.

Revised original supplementary Table 4. Summary the fitting parameters of *ex-situ* V K-edge EXFAS curves for the $\text{Ni}_3\text{Fe}_{0.5}\text{V}_{0.5}$ catalyst before and after OER at 1.75 V electrolysis.

Catalyst	Shell	Coordination number (CN)	Bond distance (R (Å))	ΔE_0 (eV)	σ^2 (10^{-3} \AA^2)	R-factor or
$\text{Ni}_3\text{Fe}_{0.5}\text{V}_{0.5}$	V-O1	4.6 ± 0.4	1.72 ± 0.01	6.4 ± 0.8	7.1 ± 0.7	0.05
	V-O2	1.4 ± 0.5	2.02 ± 0.04	6.4 ± 0.8	6.4 ± 2.3	
$\text{Ni}_3\text{Fe}_{0.5}\text{V}_{0.5}^*$	V-O1	4.4 ± 0.7	1.66 ± 0.01	-1.8 ± 1.3	2.1 ± 0.9	1.34
	V-O2	2.0 ± 1.0	2.02 ± 0.04	-1.8 ± 1.3	2.1 ± 0.9	

Note: * Data were collected after OER at 1.75 V electrolysis in 1 M KOH solution.

Question 4: “How many electrodes were prepared for each oxide? Are the measurements reproducible?”

Response 4: At least three electrodes were prepared for each hydroxide, and the spectroscopic and catalytic measurements of the hydroxide catalysts were reproducible. A short sentence to emphasize this point has been added to the part of Methods at page 24.

Question 5: “Mechanism”

Question 5a: “I like to point out that the mechanism of the LDH is still actively disused, e.g. in “onlinelibrary.wiley.com/doi/10.1002/aenm.201600621/full”. Can the author’s data support one of the previous proposals or the selection of the used reaction steps?”

Response 5a: Thanks for reminding us of this valuable review [*Adv. Energy Mater.* **6**,

1600621 (2016)]. In our DFT calculations of the oxygen binding energies and overpotentials for OER, the four single-electron-loss pathway with OH*, O* and OOH* oxygen intermediates in alkaline electrolytes was considered, which is similar to the dominant mechanism of OER for 3d metal-based (oxy)hydroxides [ACS Catal. **4**, 1148–1153 (2014); Chem. Phys. **319**, 178–184 (2005)]. The calculation results show that for Ni₃Fe_{0.5}V_{0.5} the formation of OH* and O* species is the RLS of OER process on Ni and Fe sites, whereas on V sites the formation of OOH* is the RLS. Moreover, the DFT calculations together with spectroscopic studies reveal that for Fe/V co-doped Ni (oxy)hydroxides the OER activity of doped V sites was greatly enhanced by the surrounding Ni/Fe next-nearest neighbors. This inference is agree with the statement made by Bell et al. and Calle-Vallejo et al. that for Fe-doped Ni (oxy)hydroxides the surrounding Ni next-nearest neighbors increase the activity of Fe sites [J. Am. Chem. Soc. **137**, 1305–1313 (2015); ACS Catal. **5**, 5380–5387 (2015)]. More importantly, our DFT calculations show that the (Ni₃Fe_{0.5}V_{0.5})-OOH models with some aggregated Fe and V atoms have lower formation energy and higher OER activity than the models with isolated Fe and V atoms, which indicates that the Fe neighbors near to the V play a crucial role in the enhancement of catalytic activity of the V sites in Ni₃Fe_{0.5}V_{0.5}.

The fundamental mechanism of the enhanced activity can be explained by the *d*-band theory raised by Nørskov et al. [Proc. Natl Acad. Sci. USA **108**, 937–943 (2011)]. The theory states that transition metals with higher *d*-band center result in less occupancy of the antibonding states formed with the adsorbed reaction intermediates, and thus have stronger binding capability. Consistently, our calculations on the density of states (DOS) indicate that the V atoms of Ni₃Fe_{0.5}V_{0.5} have much higher *d*-band center (0.09 eV) than Fe (–2.55 eV) and Ni (–2.78 eV) atoms, which results in less occupancy of the antibonding states of V atoms with adsorbed oxygen intermediates. As such, the V sites exhibit optimal binding capability for OER catalysis with regard to Ni and Fe atoms (Fig. 7c) and have the lowest theoretical overpotential for OER compared to the Ni and Fe sites.

Inspired by the referee’s comment, we added the following discussion at page 22 of the revised manuscript and cited two more relevant literatures:

“The DFT calculations together with spectroscopic studies reveal that for the Fe/V co-doped Ni (oxy)hydroxide the OER activity of doped V sites was greatly enhanced by the surrounding Ni/Fe next-nearest neighbors. This inference is agree with the statement made by Bell et al. and Calle-Vallejo et al. that for Fe-doped Ni (oxy)hydroxides the surrounding Ni neighbors increase the activity of Fe sites^{10,60,61}. More importantly, the

DFT calculations show that the $(\text{Ni}_3\text{Fe}_{0.5}\text{V}_{0.5})\text{-OOH}$ models with some of Fe and V atoms aggregated in the $\text{Ni}(\text{OH})_2$ lattices have lower formation energy and higher OER activity than the models with isolated Fe and V atoms, which indicates that the Fe neighbors near to the V play a crucial role in the enhancement of catalytic activity of the V sites in $\text{Ni}_3\text{Fe}_{0.5}\text{V}_{0.5}$.”

Question 5b: “*The point of zero charge (PZC) of oxides is below pH 12, so at pH 14, the equilibrium should be shifted toward *O. I disagree that it should be a possible RLS since the solution thermodynamics favor its generation. The potential of charge neutrality is also much lower than the onset of the OER on oxides.*”

Response 5b: In our work, the theoretical OER overpotential and rate-limiting step were calculated by the standard hydrogen electrode method developed by Nørskov et al. [*J. Phys. Chem. B* **108**, 17886–17892 (2004)]. It does not consider the presence of H^+ ions on material surfaces, which is rather technically challenging for density functional theory (DFT) calculations. Actually, our theoretical results are not contradict with the PZC of LDHs. On the Ni site of the Ni (oxy)hydroxide and the Fe site of the Ni_3Fe (oxy)hydroxide, formation of OH^* and O^* species are both difficult and involve large potential steps (Supplementary Fig. 29a,b). The rate-limiting step was identified as the oxygen intermediate that has the highest Gibbs free energy of formation among the four elemental steps. Even though O^* species is readily to form considering the presence of H^+ ions on the catalyst surface, the 1st reaction step of OER (i.e. formation of OH^* species) is thermodynamically unfavorable on Fe and Ni sites, which still leads to lower OER activity with regard to the V sites. Inspired by the referee’s comments, we revised the discussion on the rate-limiting step in the revised manuscript (at page 21) as follows:

“In particular, oxygen binding on the Fe and Ni sites is relatively weak, i.e. $E_{\text{OH}^*} > 1.15$ eV and $E_{\text{O}^*} - E_{\text{OH}^*} > 2.24$ eV. As a consequence, formation of OH^* and O^* species encounters large potential barriers (Supplementary Fig. 29a, b) and will limit the reaction rate of OER process.”

Question 6: “*EXAFS analysis*”

Question 6a: “*Please mention for the readers that Fig. 3 and 6 show the reduced distance. The interatomic distance is determined by the fits.*”

Response 6a: Thanks for reminding us of this point. In the revised manuscript, we have change the x -coordinate from “ R ” to “Reduced distance” in Figs 3 and 6, as well as in

Supplementary Figs 8, 11, 12, 13, and 14. A sentence has been added to the section of “EXAFS data analysis” in the Supplementary Information at page 43: “The FT curves shown in Figs 3 and 6, as well as in Supplementary Figs 8, 11, 12, 13, and 14, are not phase-shift corrected, and hence the peak position is shorter than the real interatomic distance by an amount of approximately 0.3–0.5 Å depending on the type of neighboring atoms. The accurate bond length is obtained from curve-fitting.”

Question 6b: *“Please state the details of EXAFS FT. What was the windowing function and its parameters? What K-space range was used? What R-space range was analyzed/fit? This information is necessary for reproduction of the results.”*

Response 6b: Thanks for the good suggestion. The details of EXAFS FT have been added to the section of “EXAFS data analysis” at pages 42–43 in the revised Supplementary Information.

Question 6c: *“Phases from experimental references were used. Which reference materials were used? How were they extracted. This can significantly affect the obtained fit values. Could the low V–O distance be an artefact of the experimental phase function?”*

Response 6c: Phase-shift function is very important in extracting structural parameters from EXAFS curve-fittings. Before 1990s, this function was mostly extracted from the experimental spectra of standard materials with known structure. But it suffers from many disadvantages, e.g., in many cases it is hard to find the proper standard material, and the distance determination is not very accurate. Ever since the development of the *ab initio* multiple-scattering code FEFF, it could provide a reliable and convenient theoretical standard for most applications and hence overcome the disadvantages of experimental phase-shifts. Nowadays, FEFF has been used as a widely-accepted method to generate the phase-shifts for EXAFS fittings. And we also used FEFF generated phase-shifts instead of the experimental phase-shifts in the whole paper.

For our $\text{Ni}_3\text{Fe}_{0.5}\text{V}_{0.5}$ catalyst, both Fe and Ni atoms have the similar features in the FT curves, implying the substitution of Fe for the Ni sites. As we have stated in Response to Question 3c, V atoms also occupy the Ni sites although the V–M coordination peak is weak in the FT curve at V *K*-edge. Therefore, the phase-shift functions of Fe–O, Fe–Ni, V–O, and V–Ni pairs were obtained by FEFF calculations based on the structural models of a Fe (or V) atom substituting for a Ni site in the $\text{Ni}(\text{OH})_2$ lattice. The data of $\text{Ni}_3\text{Fe}_{0.5}\text{V}_{0.5}$ were fitted again by using two V–O paths

which yield better matches between the experimental data and fitting curves. More reliable structural parameters are obtained. It helps us to confirm the significantly larger local distortion of the octahedron around V than around Fe and Ni, with the obvious V–O bond length splitting. These descriptions have also been added to the section of “EXAFS data analysis” at page 43 in the revised Supplementary Information.

Question 6d: *“How is the uncertainty in the tables of the EXAFS analysis calculated? Is it the parameter error?”*

Response 6d: Thanks a lot for the referee’s comment. The uncertainties of the best-fit parameters were estimated by the Artemis code of the Iffeffit software, which now has been used widely, and it is in line with the Standards and Criteria Committee adopted by IXS committee in the year of 2000. Based on the appropriate estimation of the uncertainty of the EXAFS measurement, the uncertainty in the table of EXAFS results estimated by Iffeffit is the error bar of the fitting parameter.

Iffeffit is as close to the “standard techniques of data fitting and error analysis” as possible. Iffeffit adopts χ^2 to judge the quality of the fit. It is a scaled measure for the sum of the squares of the difference of model and EXAFS data. The χ^2 depends on the uncertainty of the function to minimize. The EXAFS measurement uncertainty mainly comes from two aspects. One is the random fluctuations in the data, and the other is the systematic error. The random fluctuations of the data over the fit range are easy to be estimated from the r.m.s. amplitude of the *R*-space FT between 15 and 25 Å, as a single number for all data points. This assumes that the fluctuations are white noise, and that they are much bigger than the signal past 15 Å, due to the local character of the EXAFS. However, the systematic errors in the data are much more difficult to estimate. For this reason, the χ^2 estimating by Iffeffit usually does not include the systematic errors. Fortunately, the high quality of X-ray beam from the advanced SR ensures this error is very small and much steady. Therefore the systematic error may hardly influence on the comparison between local structures of different samples.

The uncertainties of the variable parameters will be estimated immediately after the best-fit values of these parameters are found, by estimating the function χ^2 . The uncertainty in the value of a variable is the amount by which χ^2 can be increased and still have a value below some limit. Generally, a common criterion has been used in Iffeffit. That is χ^2 deviates the best-fit value no more than 1. Below this limitation, the uncertainty of a variable will be calculated. In the Iffeffit, when the uncertainty in a variable is evaluated, all the other variables are allowed to vary, so that the correlations

between variables can be taken into account. A brief description on the uncertainty calculation has been added to the section of “EXAFS data analysis” at page 43 in the revised Supplementary Information.

Question 6e: “How do the fit results compare to ref. 10 and 14? This should also be included in the revision for the readers.”

Response 6e: Thanks for the advice. We have checked and compared the results in these references with ours. We find that our results of the M–O (~ 2.0 Å, M represents Fe or Ni) and (Ni/Fe)–M (~ 3.1 Å) of our as-prepared catalysts and (Ni/Fe)–O (~ 1.9 Å/1.97 Å) and (Ni/Fe)–M (~ 2.8 Å) under the activated condition are close to the results in these references. For example, the fit results are the M–O (~ 2.0 Å, M represents Fe or Ni) and (Ni/Fe)–M (~ 3.1 Å) for the resting condition and M–O (~ 1.9 Å) and (Ni/Fe)–M (~ 2.8 Å) under the active condition [ref. 10: *J. Am. Chem. Soc.* **137**, 1305–1313 (2015)], and the fit results are (Ni/Fe)–O (~ 1.9 Å/2.0 Å), (Ni/Fe)–M (~ 2.8 Å/3.0 Å) under catalytic OER conditions [ref. 14: *J. Am. Chem. Soc.* **139**, 2070–2082 (2017)].

Accordingly, we tried to make a clear statement for the readers by adding the following sentence to the revised manuscript at page 17: “These FT-EXAFS quantitative fit results of Ni and Fe *K*-edges of Ni₃Fe_{0.5}V_{0.5} catalysts in both rest and activated states are consistent with the previous reports^{10,14}.”

Question 7: “The intensity (or better area under the curve) of soft XAS spectra does depend on the number of holes but also on other factors”

Question 7a: “How were the spectra normalized?”

Response 7a: Yes, the intensity of soft XAS spectra depends on the number of holes and other factors. For annihilating the effect of different sample concentration and measurement conditions on the intensity of characteristic XAS peaks, the data at Ni, Fe and V *L*-edge were normalized following the method proposed in the literature [*Phys. Rev. B* **52**, 3143–3150 (1995)]. For example, at the Fe *L*-edge, the experimental spectra have been normalized with respect to the background; before the absorption edge, it is equal to 0 and after the absorption edge (728 eV) it is equal to 1. This point has been mentioned at page 26 in the section of “Methods”.

Question 7b: “How do the soft XAS spectra look at different positions on the sample? Are the changes significant?”

Response 7b: For optimizing the XAS measurements, we collected several XAS spectra by irradiating the soft X-ray beam at different positions on each sample. No big difference was found among these XAS spectra. This is mainly due to the uniformity of our samples, which were fine powders mixed with graphite and pressed into a 13 mm diameter pellet. The sample is much larger than the beam size of the soft X-ray ($3 \times 1 \text{ mm}^2$), guaranteeing the reliability of our soft XAS spectra. This point has been mentioned at page 26 in the section of “Methods”.

Question 7c: *“Can the observed changes due to change in the total absorption when another element is added to the oxide?”*

Response 7c: When another element is added to the oxide, the total absorption will increase, because it increases the background absorption. But this does not change the characteristic peaks of the absorbing elements (here V, Fe, and Ni), because they are solely determined by the quantity of the absorbing elements. This is why XAFS could be called an “element-specific” technique.

Question 7d: *“A significant change in holes on the element should also lead to a shift of the peak position, which is only observed for V.”*

Response 7d: Yes. The *L*-edge absorption arises from the transition of *2p* electron to the unoccupied *3d* orbitals, namely, *3d* holes. Hence, *L*-edge XAS provides a means to detect the *3d* holes. Numerous experimental and theoretical studies have shown that a significant change in *3d* holes leads to changes of both intensity and position of *L*-edge peaks. As for our *L*-edge XAS spectra of $\text{Ni}_3\text{Fe}_{0.5}\text{V}_{0.5}$, not only V edge, but also Fe and Ni edge spectra show shifts in peak positions (Fig. 4d,e,f); the shifts of the latter are less significant, but still visible.

Question 8: *“DFT”*

Question 8a: *“Which metal valences were found in the DFT calculations? Do they match the trends obtained from XAS? The DFT and XAS results could be better connected.”*

Response 8a: Unfortunately, DFT calculations cannot identify the metal valence in multi-metal materials. The variation in electron density of a metal atom with different valences is too subtle to be captured accurately by DFT. Although it is impossible to determine the exact valence for each metal in the catalyst by examining the electron density distributions, the trend of the partial charges of ~ 1.6 , 1.0 , and $0.8e$ on the V, Fe

and Ni sites, respectively, obtained from Mulliken charge analysis, is consistent with that of the valences of $V^{4+/5+}$, Fe^{3+} , and Ni^{2+} estimated on the basis of XANES and XPS. The related discussion on the connection of DFT and XAS results has been added to page 19.

Question 8b: *“Why are there different circles of the same color in Fig. 7c? What models do they belong to? Please provide a supporting table with detail or other means of documentation.”*

Response 8b: Thanks for this good suggestion. For the mono-doped or co-doped Ni (oxy)hydroxide, we constructed various models with the dopants located in different relative positions, or with the material surface covered by different oxygen species. In Fig. 7c, different circles of the same color correspond to the results from one type of reaction site (V, Fe, or Ni) in different structural models. Following the referee’s suggestion, we relates each data point in Fig. 7c to specific structural model, whose detailed information is given in Supplementary Fig. 28 and Supplementary Table 6. The original Fig. 7c has been replaced with the revised one, and the related figure caption has been changed.

Question 8c: *“Are surrounding atoms and the electrolyte considered in calculations of the oxygen adsorption? Please add detail in the main text.”*

Response 8c: Yes, we have considered Ni (oxy)hydroxide models with surfaces covered by water molecules or oxygen species (O^* , OH^*) that are possibly present in the reaction media. These models with different covered species give very similar results on the catalytic properties as illustrated by Fig. 7c and Supplementary Table 6. Following the referee’s suggestion, we added the following information to the main text (pages 18–19) of the revised manuscript:

“The model surfaces are covered by either water molecules or oxygen species that are possibly present in the reaction media. These models with different covered species give very similar results on the catalytic properties (Fig. 7c and Supplementary Table 6).”

Question 8d: *“P.19. ‘The highest activity is achieved’. This refers to a calculation and thus ‘is predicted’.”*

Response 8d: The sentence has been revised to “The highest activity is predicted on the V site of ...” at page 22 of the revised manuscript.

Question 9: “*Errors or ambiguous discussion*”

Question 9a: “*P10 ‘For Ni₃Fe (oxy)hydroxide, the unpaired electron in the π -symmetry (t_{2g}) d-orbital of Fe³⁺. The Fe-L spectrum in Fig. 4e is clearly that of high spin Fe³⁺ ($t_{2g}^3 e_g^2$). Thus, there are three unpaired t_{2g} electrons. I also recommend giving the t_{2g} and e_g occupancies instead of the d-electron count on P10, which would make it easier to follow. Lastly, the soft XAS spectra should be discussed in terms of t_{2g}/e_g occupancy or at least whether the ion is high or low spin, which can be done, e.g., using the branching ratio (L_3/L_2+L_3).’*”

Response 9a: Thanks for referee’s valuable suggestion. Following this suggestion, we calculated the branching ratio, $L_3/(L_2+L_3)$, at the Fe L-edge according to the literature [*J. Am. Chem. Soc.* **119**, 4921–4928 (1997)]. The branching ratio is approximately 0.74, implying the high-spin of Fe³⁺. Furthermore, we calculated the Fe $L_{2,3}$ -edge XAS spectra for the high-spin and low-spin model of Fe³⁺ (Supplementary Fig. 17). Obviously, the calculated high-spin $L_{2,3}$ -edge XAS spectrum could well produce the experimental data, affording more evidence for the high-spin configuration of Fe³⁺ substituting the Ni sites. The related calculation process of the $L_{2,3}$ -edge XAS theoretical spectra of high/low-spin models of Fe³⁺ has been added to “Supplementary Note 1” at page 40 and “Supplementary Methods” at pages 43–44 of the revised Supplementary Information.

Accordingly, the related discussion has been added to page 11 of the revised manuscript: “Moreover, we calculated the branching ratio, $L_3/(L_2+L_3)$, at the Fe L-edges of Ni₃Fe and Ni₃Fe_{0.5}V_{0.5}, which is approximately 0.74, implying the high-spin of Fe³⁺ (ref. 46). And we also calculated the Fe $L_{2,3}$ -edge XAS spectra for the high-spin and low-spin models of Fe³⁺ (Supplementary Note 1 and Supplementary Methods). Obviously, the calculated high-spin $L_{2,3}$ -edge XAS spectrum could well produce the experimental data (Supplementary Fig. 17), affording more evidence for the high-spin configuration of Fe³⁺ substituting the Ni sites. Thus, the valence electronic configurations of Ni²⁺, Fe³⁺, V⁴⁺ and V⁵⁺ are $3d^8 (t_{2g}^6 e_g^2)$, $3d^5 (t_{2g}^3 e_g^2)$, $3d^1 (t_{2g}^1 e_g^0)$ and $3d^0 (t_{2g}^0 e_g^0)$, respectively, which are adopted in the following analysis of valence electron structures of metal ions in Ni₃Fe, Ni₃V, and Ni₃Fe_{0.5}V_{0.5}.”

Question 9b: “*The authors use “charge transfer” frequently. It is confusing and imprecise. Which charge carrier? Electron or hole? It would be clearer to state which element would be reduced or oxidized.*”

Response 9b: Thanks for the good suggestion. The expression “charge transfer” has been revised to “partial electron transfer” in the entire manuscript to avoid the ambiguity.

Question 9c: *“P14. The ECSA was not measured. It is obtained by dividing the capacitance of the sample by that of a perfectly flat reference of the same material, see ref. 3. The used normalization corrects for difference in roughness among the samples and is thus useful but should not be called ECSA and cannot be compared to catalysts on other substrates.”*

Response 9c: According to the comment of referee #1, the ECSAs of the $\text{Ni}_3\text{Fe}_{1-x}\text{V}_x$ (oxy)hydroxide electrodes have been revised to the roughness factors (RFs), and according to the equation given in ref. 3 [*J. Am. Chem. Soc.* **135**, 16977–16987 (2013)], the J_g values were normalized by RFs to get the specific current densities (J_s). Comparison of the J_s values were made only within the CFP electrodes fabricated in the present work. The corresponding discussion at page 15 has been revised; in addition, the Supplementary Note 2 and Supplementary Fig. 23 have been revised.

Question 9d: *“Supplemental Figures S5+S6 do not show “atomic resolution”. They are blurry and individual atomic columns are not resolved.”*

Response 9d: The words “atomic resolution” have been removed from the captions of Supplementary Figs 5 and 6. The descriptions of “atomic resolution” for Supplementary Figs 5 and 6 have also been removed from the main text. Supplementary Fig. 5 just shows that single atoms, clusters, and small particles of Fe and V species are not observed in aberration-corrected high-angle annular dark-field scanning TEM (HAADF-STEM) images of $\text{Ni}_3\text{Fe}_{0.5}\text{V}_{0.5}$ NSs. And Supplementary Fig. 6 together with Fig. 2f in the manuscript provide evidence for the uniform distribution of Ni, Fe, V, and O elements in the as-prepared NSs.

Question 9e: *“Supplemental P37. NiOOH was not found by PXRD or HRTEM. It is only detected by Raman and in situ XAS.”*

Response 9e: We fully agree with referee’s comment that the NiOOH was detected by the *in-situ* Raman and XAS spectroscopy, but not by PXRD and HRTEM. However, our PRXD and HRTEM measurements on the as-prepared $\text{Ni}_3\text{Fe}_{0.5}\text{V}_{0.5}$ show that the (101) surface is the dominant exposed surface of $\text{Ni}(\text{OH})_2$. Moreover, previous theoretical and experimental studies demonstrated that the LDHs undergo phase transition to the

(oxy)hydroxide during OER process in alkaline media, as the (oxy)hydroxide phase is thermodynamically most stable under typical OER conditions [*Electrochim. Acta* **11**, 1079–1087 (1966); *Nat. Mater.* **11**, 550–557 (2012); *Phys. Chem. Chem. Phys.* **15**, 13737–13783 (2013); *Science* **345**, 1593–1596 (2014); *Nat. Commun.* **6**, 7261 (2015); *Adv. Energy Mater.* **6**, 1600621 (2016); *Nano Lett.* **16**, 7718–7725 (2016)]. Therefore, we infer that in our experiment, the OER process is most probably catalyzed by the (101) surface of NiOOH.

The corresponding sentence at page 45 of the revised Supplementary Information has been revised to “We consider the (101) surface of β -NiOOH for OER catalysis because (i) the PXRD and HRTEM experiments on the as-prepared Ni₃Fe_{0.5}V_{0.5} show that the (101) surface is the dominant exposed surface of Ni(OH)₂, (ii) previous theoretical and experimental studies demonstrated that the (oxy)hydroxide phase of LDH is the stable phase under typical OER conditions in alkaline media^{10,14,33,39}, and (iii) in our case NiOOH has been detected by *in-situ* Raman and XAS spectroscopies for the activated Ni₃Fe_{0.5}V_{0.5} catalyst.”

Question 10: “*Minor/typo*”

Question 10a: “*Supplemental Fig. 10. What does the label Ni stand for? Ni metal? Ni(OH)₂? Please add simple Ni oxide references.*”

Response 10a: The label Ni in the original Supplementary Fig. 10a stands for Ni(OH)₂. The labels “Ni” in all Figs and Tables have been revised to Ni(OH)₂.

Question 10b: “*The “Janus face property” is called amphoteric in chemistry or is something else meant?*”

Response 10b: The sentence, “the Janus-face property of Fe dopant ...” at page 10, has been revised to: “the disparate functions of Fe dopant in different Ni(OH)₂-based materials, that is, Fe dopant acts as an electron accepting site in Ni₃Fe, while it functions as an electron donating site in an integrated effect when V is co-doped with Fe into the Ni(OH)₂ lattices.”

Question 10c: “*P14. Which of the two charge transfer resistances is reduced?*”

Response 10c: Both charge transfer resistances ($R_{ct(int)}$ and $R_{ct(s-l)}$) are reduced (please see Supplementary Table 9). The following short paragraph has been added to the revised manuscript at page 16 to make this issue more clear in the main text: “The Nyquist plots (Fig. 5d) are fitted to a simplified Randles equivalent circuit model

(Supplementary Note 3). The very small semicircles in the high frequency zone are attributed to the internal charge-transfer resistances ($R_{ct(int)}$) of electrodes, and the second semicircles represent the charge-transfer resistances ($R_{ct(s-l)}$) at the solid/liquid interface between electrode and electrolyte. Both $R_{ct(int)}$ and $R_{ct(s-l)}$ values apparently decreased as Fe and V were co-doped into the Ni(OH)₂ lattices. The total charge-transfer resistances (R_{ct}) measured at 300 mV overpotential are 4.2, 7.2, 10.0, and 17.2 Ω for the CFP-supported Ni₃Fe_{0.5}V_{0.5}, Ni₃V, Ni₃Fe, and pure Ni (oxy)hydroxide catalysts, respectively (Supplementary Table 9).”

Question 10d: “Supplemental Table 9. Please add potential at which EIS was performed and refer to circuit.”

Response 10d: The applied potential ($\eta = 300$ mV) for EIS measurement has been added to the caption of Supplementary Table 9.

Question 10e: “10e) Please index PXRD patterns. Which reflections change/shift? Are those assigned to the metal oxide slabs or to the distance between the metal oxide slabs (of the LDH)?”

Response 10e: Following the suggestion of the referee, the peaks in the PXRD patterns (Supplementary Fig. 2) have been indexed. The following sentences have been added to discuss the shifts of 2θ values at page 4: “The PXRD patterns (Supplementary Fig. 2) indicate that Ni₃Fe_{1-x}V_x (oxy)hydroxides are isostructural to α -Ni(OH)₂ (JCPDS Card No. 38-0715). The reflections at $2\theta = 11.4^\circ$ and 22.7° , corresponding to the (003) and (006) lattice planes of the Ni₃Fe_{1-x}V_x (oxy)hydroxides, slightly shifts to larger 2θ values by 0.2° and 0.6° , respectively, relative to those of the α -Ni(OH)₂ reference. The d -spacing values obtained from the (003) and (006) reflections are about 7.65 and 3.81 \AA , respectively, with a small contraction compared to those for pure α -Ni(OH)₂ ($d(003) = 7.79$ \AA and $d(006) = 3.91$ \AA), which is caused by the substitution of Fe and V atoms for Ni in the lattice sites of the Ni hydroxide matrix^{32,39,40}.”

Question 10f: “I was confused by Fig. 1, where the substrate looks like a metal mesh. The carbon fiber substrate is not as ordered as shown there.”

Response 10f: The schematic diagram of the CFP substrate in Fig. 1 has been redrawn to make it look like a CFP. Besides, the figure for TOC has also been revised.

Responses to the comments of Referee #2:

Comment: *“This is a very comprehensive paper of very high scientific quality and considerable topical interest. The development of metal oxide electrocatalytic materials which catalyse the electro-oxidation of water to generate molecular oxygen is a grand challenge in energy science. Doping metal oxide lattices with small quantities of foreign metal atoms has been shown to be a useful partway to lower the overpotential required to drive the water oxidation at a fixed current density (say 10–20 mA/cm²). The process of doping nickel oxyhydroxide with iron has been shown over the last few years, to significantly enhance the catalytic activity of the bimetallic catalyst.*

In the present paper, the authors have convincingly shown that the addition of vanadium to the bimetallic catalyst can result in even better catalytic activity. An overpotential of ca. 200 mV at 10 mA/cm² with a Tafel slope of ca. 39 mV/dec has been measured for a Ni₃Fe_{0.5}V_{0.5} oxyhydroxide material. These are impressive performance indicators. This is a very significant result.

The paper is very complete. The characterization and analysis of the material and its performance under operating conditions has been rigorously examined. Furthermore the paper contains significant DFT analysis as an aid to mechanistic interpretation.

Given the quality of the science and its inherent broad interest and topicality, I strongly recommend that this paper be accepted for publication.”

Response: Thanks for the positive comments on this part of work. We will try our best to further revise the manuscript to make it suitable for publication in *Nat. Commun.*

Responses to the comments of Referee #3:

Comment: *“This manuscript aims to unveil OER improvements through Fe/V co-doping in Ni(OH)₂ by comprehensive spectroscopic studies including XAFS, sXAS and Raman along with advanced TEM of catalysts ex-situ and in-situ of OER. The experimental results have been combined with calculation to get a conclusion that V site is the active sites in this catalysts. the methodology is appealing and should deliver new scientific insights for this important catalyst. unfortunately the current version doesn't get there. therefore I recommend a major revision plus another round review. in the revision the author needs to carefully address those questions:”*

Question 1: *“V in situ XAFS needs to be conducted since this is the proposed active*

sites. I understand that in situ work for V is a bit challenge but it is still doable.”

Response 1: Thanks for the suggestion. We agree well with the referee’s comment that the *in-situ* V *K*-edge XAS experiment is very important for understanding the active sites and reaction mechanism. So again, we performed the *in-situ* experiments of V *K*-edge in the fluorescence mode using a Lytle detector on beamline BL14W1 at Shanghai Synchrotron Radiation facility (SSRF, China).

Interestingly, the *in-situ* V *K*-edge XANES spectra displayed that the intensity of the pre-edge peak was gradually decreased as the applied potential was increased from 1.15 to 1.75 V (inset of Fig. 6f), however, it shows identical spectral features (including shape and intensity of other characteristic peaks) to those measured before OER. Similarly, except for the decrease in the intensity of the characteristic peaks, no other obvious change is visible at the *ex-situ* V *L*-edge spectra (Supplementary Fig. 25) after OER measurement at 1.75 V. This evidence suggests partial electron transfer to the V *3d* orbitals, as their peak intensity is proportional to the unoccupied density of *3d* states. Meanwhile, the V–O distance is also shrunk from 1.70 Å at 1.15 V to 1.65 Å at 1.75 V (Fig. 6g and Supplementary Table 8), and the V atoms with such a short V–O bond may have optimal binding capability with oxygen intermediates relative to Ni and Fe atoms, and exhibit enhanced OER activity.

The related discussion on the results of *in-situ* V *K*-edge XAS experiment has been added in the revised manuscript at pages 17–18 of the revised manuscript.

Question 2: *“from in situ XAFS, the structure of Ni/Fe in the active catalyst under OER shall be resolved; therefore this structure information shall be used in calculation rather than using models from literature.”*

Response 2: Actually, our models for DFT calculations were constructed based on the *in-situ* XAFS experimental results of Ni/Fe *K*-edge. In the models, the coordination number of the metal center (Ni, Fe, and V) is about 6, close to the EXFAS fit results (Supplementary Table 8). The optimized average bond lengths for Ni–O and Fe–O in the resting hydroxide phase are 2.01 and 1.97 Å, respectively, and in the active (oxy)hydroxide phase are 1.97 and 1.93 Å, which are in good agreement with the *in-situ* XAFS experimental values (2.04 ± 0.02 Å (Ni–O) and 2.00 ± 0.01 Å (Fe–O) in the resting state; 1.90 ± 0.04 Å (Ni–O) and 1.97 ± 0.01 Å (Fe–O) in the active state). The small deviations of the theoretical data from the XAFS experimental values may be ascribed to the presence of defects in the synthetic samples, which were not considered in our models. Nevertheless, our DFT calculations clearly show the theoretical trend of

OER activity for the V, Fe, and Ni sites, and elucidated the intrinsic relation between electronic structure and OER performance of multimetal (oxy)hydroxide catalysts.

Question 3: “the manuscript only “proves” the knowing knowledge that V is the active sites from calculation (optimal binding energy for OER intermediates at V site) and XAFS (ultra short V-O bond). a more deep discussion that how such local structural change led to the calculated results are need and this probably unveil atomic level new insights of the high OER catalysis in this known catalyst.”

Response 3: Thanks for referee’s suggestion, which helps us improve the manuscript. To the best of our knowledge, to date, there is no report on in-depth spectroscopic and theoretical studies of local coordination environment, electronic band structure and active site for the V-containing trimetal (oxy)hydroxide OER catalysts in both rest and activated states, and only two very recent reports [*Nat. Commun.* **7**, 11981 (2016); *Angew. Chem. Int. Ed.* **56**, 3289–3293 (2017)] indicate that the doped V is the possible active site in V-containing bimetal (oxy)hydroxide OER catalysts on the basis of DFT calculations. The spectroscopic (XAFS/XPS) results and theoretical calculations of this work provide some fundamental insights into the V/Fe co-doped Ni (oxy)hydroxide OER catalysts. In general, the results obtained from the comparative studies of the V-containing bi- and trimetal (oxy)hydroxide OER catalysts by soft XAS/XPS provide adequate evidence for the synergetic interaction of the V/Fe dopants and the Ni host in the Fe/V co-doped Ni (oxy)hydroxides, and demonstrate the subtle modulation of the local coordination environment and electronic structure of the Fe/V/Ni cations by co-doping Fe/V into the Ni(OH)₂ lattices. The *in-situ* XAS analyses manifest for the first time the contraction of M–M' and M(M')–O bond lengths and the short V–O bond distances in the activated V/Fe co-doped Ni (oxy)hydroxide catalyst. Accordingly, the bond lengths between metal-oxygen intermediates in our DFT models are 1.60–1.84, 1.63–1.95 and 1.77–2.05 Å for V, Fe, and Ni, respectively, which are in good agreement with the trend of experimental XAS results. The distinct bond length between various transition metals and O atom is fundamentally governed by the electronic band structure of the material. Our calculations show that the V, Fe, and Ni atoms in the co-doped Ni (oxy)hydroxide have the *d*-band center of 0.09, –2.55, and –2.78 eV, respectively, as revealed by the density of states in Fig. 7d. According to the *d*-band theory [*Proc. Natl Acad. Sci. USA* **108**, 937–943 (2011)], transition metals with higher *d*-band center possess less occupancy of the antibonding states formed with adsorbates and have stronger binding strength. Thus, the bond order and binding

strength follow the sequence: V–O > Fe–O > Ni–O. Consistent with this inference, our DFT calculations show that the V site gives near-optimal binding energies of OER intermediates and has lower overpotential compared with Ni and Fe sites in the Fe/V co-doped Ni (oxy)hydroxides. To our knowledge, this is the first in-depth spectroscopic study combined with theoretical calculation on local coordination environment and electronic structure for the V-containing bi- and trimetal (oxy)hydroxide OER catalysts in both rest and activated states, and the first time to spectroscopically demonstrate the short V–O bond distances in the activated V-containing trimetal (oxy)hydroxide OER catalysts.

Following the referee’s suggestion, we added some additional discussion on the relations of high OER activity of the Fe/V co-doped Ni (oxy)hydroxide with its local structural change and electronic band structure in the revised manuscript.

At pages 17–18: “Meanwhile, the V–O1 distance is also shrunk from 1.70 Å at 1.15 V to 1.65 Å at 1.75 V (Fig. 6g and Supplementary Table 8), which is close to that of the shortest V–O bond length reported for V⁵⁺ oxides (1.59 Å) while much shorter than that reported for V⁴⁺ oxides (1.76 Å)⁴². The V atoms with such a short V–O bond may have optimal binding capability with oxygen intermediates relative to Ni and Fe atoms, and exhibit enhanced OER activity, as will be illustrated by following theoretical calculations. These *in-situ* XAS analyses manifest for the first time the contraction of M–M' and M(M')–O bond lengths and the short V–O bond distance in the activated V-containing (oxy)hydroxide OER catalysts.”

At pages 19: “In the optimized models, the bond lengths between metals and oxygen intermediates are 1.60–1.84, 1.63–1.95, and 1.77–2.05 Å for V, Fe, and Ni, respectively, which are in good agreement with the trend of experimental XAS results. The distinct bond length between O atom and V, Fe, or Ni element is a reflection of their different bond order and bond strength, which is fundamentally governed by the electronic band structure of the material. As revealed by the density of states (DOS) in Fig. 7d, the V, Fe, and Ni atoms in the co-doped Ni (oxy)hydroxide have the *d*-band center of 0.09, –2.55, and –2.78 eV, respectively. On the basis of the *d*-band theory⁵⁴, the V atoms with higher *d*-band center possess less occupancy of the antibonding states with adsorbed oxygen intermediates, and thus exhibit enhanced oxygen binding with regard to Ni and Fe atoms (Fig. 7c).”

Question 4: “*in addition to those above main concerns the claim of a uniform substitution of Fe/V into Ni(OH)₂ lattice to substitute Ni sites cannot be supported by*

TEM-EDX mapping. EXAFS at V edge also seems against the claim.”

Response 4: Yes, EXAFS at Fe/V *K*-edge is hard to definitely determine if the substitutional Fe/V atoms are *uniformly* distributed in the Ni(OH)₂ host at the present stage. Nevertheless, it gives convincing evidence for substitutionally doping of Fe/V atoms into the Ni(OH)₂ lattices and for excluding the formation of separate Fe- and V-hydroxide phases in the co-doped Ni(OH)₂ material.

The related sentence has been revised to: “both the EDX elemental mappings and linear scanning compositional analysis of the HAADF-STEM image of Ni₃Fe_{0.5}V_{0.5} NSs with sub-nanometer resolution (Fig. 2f and Supplementary Fig. 6) provide direct-viewing evidence for the uniform distribution of Ni, Fe, V, and O elements in the as-prepared NSs.”

We are very grateful to the referees for their professional comments and constructive suggestions to help us improve the quality of this manuscript.

REVIEWERS' COMMENTS:

Reviewer #1 (Remarks to the Author):

The extensive changes and additional data provided by the authors significantly improved the manuscript for the readers. Moreover, my concerns regarding novelty, synergy and the mechanism have been well addressed. The assignment of the vanadium species was clarified with additional analyses. I now find all discussions and conclusions very convincing and fully support its publication in Nature Communications because the authors' work shows that V-doping can further enhance the activity without compromising the stability of the material (and composite electrode). As such, the work should gain the attention of the electrocatalysis and solar fuels communities.

Reviewer #3 (Remarks to the Author):

The author has carefully addressed all concerns. The revision is very solid and should be published as is.

Dear editor,

The following is the comments from referees in the second run and our response.

Comments from referee #1:

“The extensive changes and additional data provided by the authors significantly improved the manuscript for the readers. Moreover, my concerns regarding novelty, synergy and the mechanism have been well addressed. The assignment of the vanadium species was clarified with additional analyses. I now find all discussions and conclusions very convincing and fully support its publication in Nature Communications because the authors’ work shows that V-doping can further enhance the activity without compromising the stability of the material (and composite electrode). As such, the work should gain the attention of the electrocatalysis and solar fuels communities.”

Comments from referee #3:

“The author has carefully addressed all concerns. The revision is very solid and should be published as is.”

Response:

There are no more issues raised by the referees in the second run. We are very grateful to all referees for their precious time spend on considering our manuscript. Their professional comments and constructive suggestions really helped us to have improved

the quality of this manuscript and finally to make it accepted by Nat. Commun.